# Finite-Time Analysis of Adaptive Temporal Difference Learning with Deep Neural Networks[*]

**Tao Sun**
College of Computer
National University of Defense Technology
Changsha, Hunan 410073, China
nudtsuntao@163.com

**Dongsheng Li**[♯]
College of Computer
National University of Defense Technology
Changsha, Hunan 410073, China
dsli@nudt.edu.cn

**Bao Wang**[♯]
Scientific Computing & Imaging Institute
University of Utah, USA
wangbaonj@gmail.com

## Abstract

Temporal difference (TD) learning with function approximations (linear functions or neural networks) has achieved remarkable empirical success, giving impetus to the development of finite-time analysis. As an accelerated version of TD, the adaptive TD has been proposed and proved to enjoy finite-time convergence under the linear function approximation. Existing numerical results have demonstrated the superiority of adaptive algorithms to vanilla ones. Nevertheless, the performance guarantee of adaptive TD with neural network approximation remains widely unknown. This paper establishes the finite-time analysis for the adaptive TD with multi-layer ReLU networks approximation whose samples are generated from a Markov decision process. Our established theory shows that if the width of the deep neural network is large enough, the adaptive TD using neural network approximation can find the (optimal) value function with high probabilities under the same iteration complexity as TD in general cases. Furthermore, we show that the adaptive TD using neural network approximation, with the same width and searching area, can achieve theoretical acceleration when the stochastic semi-gradients decay fast.

## 1 Introduction

Temporal difference (TD) learning is a popular and successful iterative algorithm in the area of reinforcement learning (RL) to evaluate a given policy [47, 51], often employed for critic part evaluation in various RL algorithms [28, 41, 42]. Classical TD algorithm adopts the tabular representation for the value function, which stores value estimates on a per-state basis. In large-scale scenarios, the tabular approach becomes intractable due to a large number of states. Thus, the function approximation techniques have been integrated with TD for better scalability and efficiency [4, 50, 36, 57]. The function approximation techniques have achieved remarkable empirical success and are theoretically justifiable when the linear function approximation is used [23]. As a special function approximation approach, deep neural networks (DNNs) have also been integrated with TD [38, 37, 33], achieving

---

[*]This work is sponsored in part by National Key R&D Program of China (2021YFB0301200), Hunan Provincial Natural Science Foundation of China (2022JJ10065), and the National Science Foundation of China (62025208 and 61906200).

36th Conference on Neural Information Processing Systems (NeurIPS 2022).

phenomenal performance in several applications. However, from the theoretical perspective, establishing theoretical convergence guarantees for training DNNs is much more complicated than that for the linear approximation algorithms, which is still widely open. Some convergence results of TD with DNN approximations have been proved by the authors of [21, 7, 53], under some extra assumptions and restrictions. The adaptive methods for DQN have been proposed in [37, 22], inspired by the adaptive stochastic algorithms to accelerate TD. The numerical results show that the adaptive versions of TD can outperform the vanilla ones in many tasks.

Mathematically, the (projected) TD (with function approximation) can be described as

$$\boldsymbol{\theta}^{k+1} = \mathbf{Proj_V}(\boldsymbol{\theta}^k - \eta \mathbf{g}^k), \tag{1}$$

where $\mathbf{V}$ is the constrained set, and $\mathbf{Proj_V}$ denotes the projection onto $\mathbf{V}$, $\boldsymbol{\theta}^k$ is the iterate, and $\mathbf{g}^k$ is the stochastic semi-gradient [2]. Although the update scheme of TD looks similar to the stochastic gradient descent (SGD), it is much more complicated due to the Markov noise, even in the linear function approximation case. Motivated by the adaptive SGD [20, 27], the adaptive TD with linear function approximation is proposed in [52, 46]. The main difference between the adaptive TD and the standard TD lies in the use of adaptive stepsize and momentum. This paper considers the neural adaptive TD. In the $k$th iteration of neural adaptive TD, it performs

$$\begin{cases} \mathbf{m}^k = \beta \mathbf{m}^{k-1} + (1 - \beta) \mathbf{g}^k, \\ v^k = v^{k-1} + \|\mathbf{g}^k\|^2, \\ \boldsymbol{\theta}^{k+1} = \mathbf{Proj_V}(\boldsymbol{\theta}^k - \eta \mathbf{m}^k / (v^k)^{\frac{1}{2}}), \end{cases} \tag{2}$$

where $\beta > 0$ is the momentum parameter, and $\eta > 0$. The points $\mathbf{m}^k$ and $v^k$ contain past information. The numerical results have demonstrated the advantage of adaptive TD over vanilla TD. The adaptive TD is proved to be convergent with linear function approximation, but the convergence remains unclear when the neural network approximation is used.

*This paper proves that the adaptive TD with neural network approximation converges when the width of a ReLU network is sufficiently large. Moreover, we prove that adaptive TD is faster than TD with the ReLU DNN approximation.*

## 1.1 Related Works

**Analysis and the recent development of TD.** Leveraging the stochastic approximation techniques, the authors in [25] establish the first convergence results for TD. The limiting convergence of TD with linear function approximation is proved in [50] with the perspective of the dynamics. Since the seminal work of [50], many works have been using the ODE-based method to study the asymptotic convergence of TD since TD update does not follow the (stochastic) gradient direction of any objective function [6, 49]. Some variants of TD have been proposed in [14] with asymptotic convergence guarantees. The first non-asymptotic analysis for the gradient TD, a variant of the TD, has been studied in [34]. Finite-time analysis of TD with independent and identically distributed (i.i.d.) observation assumption has been presented in [13]. The Markov sampling convergence analysis is proved in a subsequent paper [5]. In a concurrent line of research, TD has been studied from the perspective of stochastic linear systems [29]. The finite-time analysis for Markov sampling stochastic linear system has been developed by the authors of [43, 24]. The finite-time analysis of multi-agent TD is proved in [15]. A unified analysis for a class of TD learning algorithms with Markov jump is established in [24]. Based on Nesterov's acceleration method, a class of accelerated TD is developed and analyzed in [16]. From the algorithmic viewpoint, the adaptive TD has been recently proposed in [46, 52] to accelerate TD, inspired by the adaptive SGD. In [44], the authors present the finite-time convergence results of decentralized TD with linear approximations.

**TD with deep learning.** In contrast to the tremendous empirical success of the deep Q-networks (DQNs), the theory is still relatively weak; until recently, only a handful of papers have studied the theoretical results of TD using neural network approximation. In [21], the authors prove the convergence rates of fitting Q-iteration with a sparse multi-layer ReLU network under i.i.d. observations.

---

[2]We call it as semi-gradient because its stationary expectation is not the gradient of any fixed objective function.

The convergence of TD with two-layer neural network approximation is provided by [7] with i.i.d assumption on the samples. The TD-based algorithm with multi-layer ReLU networks under Markov samples is studied in [53]. The theory of the TD with a multi-layer ReLU network relies heavily on the existing results about overparameterized deep networks [26, 12, 19, 2, 1, 3, 56].

## 1.2 Difference Between our Work and Existing Works, and Technical Challenges

Existing related works contain two categories: adaptive TD with linear approximations and neural TD. However, our work is significantly different from these related works. 1) In contrast to adaptive TD with linear approximations, we consider the neural network approximation, in which case we do not have nice properties that linear approximation enjoys, and we have to consider the neural tangent kernel (NTK) regime and develop a new analysis leveraging the semi-Lipschitz continuity property. 2) Compared to neural TD, we use the adaptive stepsize and momentum, which has never been considered in neural TD.

## 1.3 Our Contributions

In this paper, we consider the adaptive TD with a multi-layer ReLU network approximation under the Markov observations. In contrast to the existing works [21, 7, 53], we study the adaptive variant of TD. The scheme of the algorithm is much more complicated, raising tremendous challenges in theoretical analysis. Our main theoretical contributions are summarized below:

- We extend the analysis of adaptive TD with linear function approximation [46, 52] to multi-layer neural network approximation under Markovian samplings. The theoretical results show that adaptive TD still works for the neural network approximation, even with deep neural networks.

- We establish the finite-time analyses of adaptive TD with multi-layer ReLU network approximation under Markov observations. In particular, we show that the adaptive algorithms guarantee convergence when the neural network is sufficiently wide, and adaptive TD with neural network approximation converges to a projected optimal action-value function. The technique required to connect Adam-type algorithms and neural TD is non-trivial since they belong to two very different research areas. To this end, we develop a new technique that uses expectation with a fixed delay, which is different from the coupling technique used by the authors of [5, 53].

- We prove that the speed of the adaptive TD can be faster than the vanilla ones with multi-layer ReLU network approximation. Specifically, we show that the adaptive ones use fewer iterations to reach the same desired error with the same network widths and searching areas. Our theoretical results first explained why Adam-type algorithms perform better than TD in DQN, which has been observed in practice.

## 2 Preliminaries

We introduce the notation, some basic concepts, and properties of TD in this section.

**Notation:** We use $\mathbb{E}[\cdot]$ to denote the expectation with respect to the underlying probability space *without* stochasticity of the initial point, and we use $\|\cdot\|$ to denote the Frobenius norm. $\sigma(\cdot)$ denotes the ReLU activation function. We use $\phi : \mathcal{S} \times \mathcal{A} \to \mathbb{R}^d$ to denote the feature map. Given a closed set $\mathbf{V}$, $\mathbf{Proj_V}(\mathbf{x})$ represents the projection of the vector $\mathbf{x}$ onto $\mathbf{V}$. The initial point is defined as $\boldsymbol{\theta}^{\text{init}}$. $\mathbf{B}(\boldsymbol{\theta}, \omega)$ denotes the ball centred at $\boldsymbol{\theta}$ with radius $\omega$. We use $a_k = \tilde{\mathcal{O}}(b_k)$ to hide the logarithmic factor of $b_k$ still with the same order. We write $a_k = \Theta(b_k)$ if $a_k = \mathcal{O}(b_k)$ and $b_k = \mathcal{O}(a_k)$, and we use $a_k = \tilde{\Theta}(b_k)$ to hide the logarithmic factor. We denote $\chi^k$ as the sub-algebra that generated by $\boldsymbol{\theta}^0, \boldsymbol{\theta}^1, \ldots, \boldsymbol{\theta}^k$, where $\boldsymbol{\theta}^k$ is the value in the $k$th iteration.

## 2.1 Markov Decision Process

For the sake of presentation, we consider the finite state space[3]. Consider a Markov decision process (MDP) described as a tuple $(\mathcal{S}, \mathcal{A}, \mathcal{P}, \mathcal{R}, \gamma)$, where $\mathcal{S}$ denotes the state space, $\mathcal{A}$ denotes the action

---

[3]Our results can be extended to infinite state cases, and we consider the finite state for simplicity.

space, $\mathcal{P}_a$ represents the transition matrix associated with action $a$, and $0 < \gamma < 1$ is the discount factor. In this case, let $\mathcal{P}_a(s'|s)$ denote the transition probability from state $s$ to state $s'$ under the action $a$. The corresponding transition reward is $r(s, a)$. We consider the stochastic policy $\pi : \mathcal{S} \to \Delta(\mathcal{A})$ that specifies a probability density of all actions given the current state $s$. $\pi(s, a)$ denotes the probability to choose action $a$ when the current state is $s$, and $\sum_{a \in \mathcal{A}} \pi(s, a) = 1$. We consider the *on policy* setting, where both target and behavior policies are $\pi$. The corresponding action-value function $\mathbf{Q}_\pi(s, a) : \mathcal{S} \times \mathcal{A} \to \mathbb{R}$ is defined as

$$\mathbf{Q}_\pi(s, a) := \mathbb{E}_\pi \Big[ \sum_{t=0}^\infty \gamma^t r(s_t, a_t) | s_0 = s, a_0 = a \Big],$$

and the associated value function $\mathbf{V}_\pi : \mathcal{S} \to \mathbb{R}$ is defined as

$$\mathbf{V}_\pi(s) := \mathbb{E}_\pi \Big[ \sum_{t=0}^\infty \gamma^t r(s_t, a_t) | s_0 = s \Big] = \sum_a \pi(s, a) \mathbf{Q}_\pi(s, a).$$

It is evident that the restriction on discount $0 < \gamma < 1$ can guarantee the boundedness of $\mathbf{Q}_\pi(s, a)$. The Markovian property of MDP yields the following celebrated Bellman equation

$$\mathcal{T}_\pi \mathbf{Q}_\pi = \mathbf{Q}_\pi, \tag{3}$$

where the Bellman operator $\mathcal{T}_\pi$ on a value function $\mathbf{Q}_\pi$ can be represented as

$$(\mathcal{T}_\pi \mathbf{Q}_\pi)(s, a) = r(s, a) + \gamma \sum_{s' \in \mathcal{S}} \mathcal{P}_a(s'|s) \mathbf{V}_\pi(s')$$

$$= r(s, a) + \gamma \sum_{s' \in \mathcal{S}} \mathcal{P}_a(s'|s) \sum_{a'} \pi(s', a') \mathbf{Q}_\pi(s', a'),$$

where $s \in \mathcal{S}, a \in \mathcal{A}$. Directly solving (3) needs $\mathcal{O}(|\mathcal{S}|^3 |\mathcal{A}|^3)$ computational cost, which is very expensive since $|\mathcal{S}||\mathcal{A}|$ is usually very large. Thus, people turn to use a linear or non-linear approximation as $\mathbf{Q}_\pi \approx \Phi(\boldsymbol{\theta}) : \mathbb{R}^D \to \mathbb{R}^{|\mathcal{S}||\mathcal{A}|}$, where $\boldsymbol{\theta} \in \mathbb{R}^D$ and $D \ll |\mathcal{S}||\mathcal{A}|$. In this way, the dimension can be significantly reduced, and we can get an approximate solution very efficiently.

We collect necessary assumptions for MDP below.

**Assumption 1** *The transition rewards are uniformly bounded by 1, that is, $|r(s, a)| \leq 1, s \in \mathcal{S}, a \in \mathcal{A}$. For any $s \in \mathcal{A}$, it holds $\mu(s) = \lim_{t \to \infty} \mathcal{P}(s_t = s | s_0 = s', a_0 = a') > 0$. There exist constants $0 \leq \rho < 1$ and $\bar\kappa > 0$ such that*

$$\sum_{s \in \mathcal{S}} |\mathcal{P}(s_t = s | s_0 = s', a_0 = a') - \mu(s)| \leq \bar\kappa \rho^t.$$

The boundedness of $(r(s, s'))_{s,s' \in \mathcal{S}}$ comes from the finiteness of $\mathcal{S}$. In Assumption 1, the uniform boundedness assumption can be replaced by non-uniform boundedness in the finite state case. In this paper, it is used for simplicity. The rest part of Assumption 1 is standard for the Markovian property. It is well-known that irreducible and aperiodic Markov chains can always follow Assumption 1 [30]. For Assumption 1, the time that $(s_t, a_t)_{t \geq 0}$ needed for its current state distribution to match the stationary one with $\epsilon$ error in total variation distance is $\mathcal{O}(\log \frac{1}{\epsilon} / \log \frac{1}{\rho})$. Thus, the constant $\rho$ represents the speed of the process accessing to the stationary distribution. In finite-time cases, it is easy to prove that $\rho$ is the second largest eigenvalue of $\mathcal{P}$. We can see that the smaller $\rho$ is, the faster the process will converge to the stationary states.

## 2.2 Neural Temporal Difference Learning

Although we can get $\mathbf{Q}_\pi$ by solving the Bellman equation induced by the given policy $\pi$, in practice $\mathcal{S}$ may contain a very large number of different states and actions, and it is hard to solve the Bellman equation directly. Thus, alternative methods have been proposed, including leveraging the linear [48] or non-linear approximations (e.g., kernel methods and neural networks [38]). This paper is devoted to the study of the approximation using a $L$-hidden-layer ReLU neural network defined as

$$f(\boldsymbol{\theta}; \mathbf{x}) = \sqrt{m} \mathbf{W}_L \sigma(\mathbf{W}_{L-1} \cdots \sigma(\mathbf{W}_1 \mathbf{x}) \cdots),$$

where $\mathbf{x} \in \mathbb{R}^d$ is the input data, $\mathbf{W}_1 \in \mathbb{R}^{m \times d}$, $\mathbf{W}_L \in \mathbb{R}^{1 \times m}$ and $\mathbf{W}_l \in \mathbb{R}^{m \times m}$ for $l = 2, \ldots, L-1$, and $\boldsymbol{\theta} := [\mathbf{W}_1, \ldots, \mathbf{W}_L]$ denotes all the weights. In neural TD, we use the following approximation

$$\mathbf{Q}_\pi(s, a) \approx f(\boldsymbol{\theta}; \phi(s, a)) = \sqrt{m} \mathbf{W}_L \sigma(\mathbf{W}_{L-1} \cdots \sigma(\mathbf{W}_1 \phi(s, a)) \cdots),$$

where $\boldsymbol{\theta} \in \mathbb{R}^{(L-2)m^2 + md + m}$ is the parameter vector, and $m$ is usually set such that $(L-2)m^2 + md + m$ is smaller than $|\mathcal{S}||\mathcal{A}|$ to reduce the difficulties caused by the high dimensionality. It is worth mentioning that factor $\sqrt{m}$ is multiplied to guarantee the output to be meaningful: in [Lemma 4.4, [8]], it is proved that $f(\boldsymbol{\theta}; \mathbf{x}) = \tilde{\mathcal{O}}(1)$ as $m$ is large and $\boldsymbol{\theta}$ is randomly initialized. If we remove $\sqrt{m}$, the function value is then of the order $\tilde{O}(1/\sqrt{m})$, which tends to $0$ as $m$ is large. We stress that such a use is standard in ReLU network theory.

**Assumption 2** *For any state-action pair $(s, a) \in \mathcal{S} \times \mathcal{A}$, we assume the feature vector is uniformly bounded such that $\|\phi(s, a)\| = 1$.*

A simple normalization can make Assumption 2 hold. This assumption is used to simplify coefficients in the subsequent proofs.

Next, we present the scheme of neural temporal difference learning. With $(s_k, a_k)_{k \geq 0}$ being a trajectory sampled from $\pi$ and a hyper-parameter $\eta > 0$, the neural TD updates with $\mathbf{g}^k \leftarrow \overline{\mathbf{g}}(\boldsymbol{\theta}^k; s_k, a_k, s_{k+1}, a_{k+1})$ in (1), where the stochastic semi-gradient is defined as

$$\begin{aligned} \overline{\mathbf{g}}(\boldsymbol{\theta}; s_k, a_k, s_{k+1}, a_{k+1}) := {}& \nabla_{\boldsymbol{\theta}} f(\boldsymbol{\theta}; \phi(s_k, a_k) \\ & \times [f(\boldsymbol{\theta}; \phi(s_k, a_k)) - r(s_k, a_k) - \gamma f(\boldsymbol{\theta}; \phi(s_{k+1}, a_{k+1}))]. \end{aligned} \tag{4}$$

The projection in scheme (2) is used to ensure the boundedness of $\boldsymbol{\theta}^k$ and simplify the convergence analysis. In the deep neural network approximation, the searching area $\mathbf{V}$ is chosen as a ball around the initialization $\boldsymbol{\theta}^{\text{init}} = [\mathbf{W}_1^{\text{init}}, \ldots, \mathbf{W}_L^{\text{init}}]$ for the ease of analysis. The detailed expression of the set $\mathbf{V}$ is given as follows:

$$\mathbf{V} := \{\boldsymbol{\theta} = [\mathbf{W}_1, \ldots, \mathbf{W}_L] \big| \|\mathbf{W}_l - \mathbf{W}_l^{\text{init}}\| \leq \omega\}, \tag{5}$$

where $1 \leq l \leq L$. It is assumed that $\boldsymbol{\theta}^{\text{init}}$ is chosen randomly from the Gaussian distribution, i.e., $\mathbf{W}_l^{\text{init}}$ is drawn from $\mathcal{N}(0, 1/m)$ with $l = 1, \ldots, m$.

The collection of all local linearization of $f(\boldsymbol{\theta}; \phi(s, a))$ at the initial point $\boldsymbol{\theta}^{\text{init}}$ is defined as

$$\mathcal{F}_{\mathbf{V}, m} := \{f(\boldsymbol{\theta}^{\text{init}}; \phi(s, a)) + \langle \nabla_{\boldsymbol{\theta}} f(\boldsymbol{\theta}^{\text{init}}; \phi(s, a)), \boldsymbol{\theta} - \boldsymbol{\theta}^{\text{init}} \rangle : \boldsymbol{\theta} \in \mathbf{V}\}.$$

If the function $f$ is linear, i.e.,

$$f(\boldsymbol{\theta}; \phi(s, a)) = \langle \boldsymbol{\theta}, \phi(s, a) \rangle, \tag{6}$$

we can see that $\mathcal{F}_{\mathbf{V}, m} = \{\langle \boldsymbol{\theta}, \phi(s, a) \rangle : \boldsymbol{\theta} \in \mathbf{V}\}$. The approximate stationary point of neural TD associated with $\mathcal{F}_{\mathbf{V}, m}$ is defined as follows.

**Definition 1 ([7])** *A point $\boldsymbol{\theta}^* \in \mathbf{V}$ is said to be the approximate stationary point if*

$$\langle \mathbf{h}(\boldsymbol{\theta}^*), \boldsymbol{\theta} - \boldsymbol{\theta}^* \rangle \geq 0, \quad \forall \boldsymbol{\theta} \in \mathbf{V}, \tag{7}$$

*where*

$$\mathbf{h}(\boldsymbol{\theta}) := \mathbb{E}\left[\hat{\Delta}(\boldsymbol{\theta}) \nabla_{\boldsymbol{\theta}} \hat{f}(\boldsymbol{\theta}; \phi(s, a))\right],$$

*and the temporal difference error $\hat{\Delta}$ is defined as*

$$\hat{\Delta}(\boldsymbol{\theta}) = \hat{\Delta}(s, a, s', a'; \boldsymbol{\theta}) := \hat{f}(\boldsymbol{\theta}; \phi(s, a)) - r(s, a) - \gamma \hat{f}(\boldsymbol{\theta}; \phi(s', a')), \tag{8}$$

*and $\hat{f}(\boldsymbol{\theta}; \phi(s, a)) \in \mathcal{F}_{\mathbf{V}, m}$.*

The convergence of neural TD is dependent on $\boldsymbol{\theta}^*$, and so is our result. In [7], it has been proved that such a definition is well-defined; the approximate stationary point exists, minimizing the mean squared projected Bellman error (MSPBE). Such a fact is more straightforward if the function $f$ is linear: the approximate stationary point of TD is identical to the unique solution to the projected Bellman equation [50].

**Algorithm 1** Neural Adaptive Temporal Difference
___
**Require: Parameters**: $\eta, \gamma, \beta, L, m, \omega$
    **Initialization**: $\mathbf{g}^0 = \mathbf{0}$, $\mathbf{m}^0 = \mathbf{0}$, $v^0 = 0$, $\mathbf{W}_l^{\text{init}} \sim \mathcal{N}(0, 1/m)$ with $l = 1, \ldots, L$, $\mathbf{V}$ is set as (5)
    **For** $k = 1, 2, \ldots$
        1.sample the trajectory $s_0, a_0, s_1, a_1, \ldots$ from $\pi$
        2.calculate $\mathbf{g}^k = \overline{\mathbf{g}}(\boldsymbol{\theta}^k; s_k, a_k, s_{k+1}, a_{k+1})$ as (4)
        3.update the parameter $\boldsymbol{\theta}^k$ as (2)
    **End for**
___

## 2.3 Neural Adaptive Temporal Difference Learning

This paper considers the neural adaptive TD. In the $k$th iteration of neural adaptive TD, we sample $\mathbf{g}^k = \overline{\mathbf{g}}(\boldsymbol{\theta}^k; s_k, a_k, s_{k+1}, a_{k+1})$ in (2) with stochastic semi-gradient (4). The set $\mathbf{V}$ is chosen as (5) with $\boldsymbol{\theta}^{\text{init}}$ also being sampled from the Gaussian distribution.

## 2.4 Some Useful Properties

For neural adaptive TD, several bounds and properties are presented in the following lemma (c.f. Lemma 6.1,[53]; Lemma 4.4, [8])

**Lemma 1** *Assume that* $(\boldsymbol{\theta}^k)_{1 \le k \le K} \in \boldsymbol{B}(\boldsymbol{\theta}^*, \omega)$, *given any* $0 < \delta < 1$ *and* $K \in \mathbb{Z}^+$, $m, \omega$ *and* $L$ *satisfying*

$$m \ge C \max\{dL^2 \log \frac{m}{\delta}, \omega^{-4/3} L^{-8/3} \log \frac{m}{\omega\delta}\}, \quad \frac{C_1 d^{3/2}}{Lm^{3/4}} \le \omega \le \frac{C_2}{L^6 (\log m)^3}, \qquad (9)$$

*then it holds*

$$\|\nabla_{\boldsymbol{\theta}} f(\boldsymbol{\theta}; \phi(s, a))\| \le C_3 \sqrt{m}, \forall s \in \mathcal{S}, a \in \mathcal{A}, \qquad (10)$$

*and*

$$|f(\boldsymbol{\theta}^k; \mathbf{x})| \le C_4 \sqrt{\log(K/\delta)}, 1 \le k \le K. \qquad (11)$$

*Denote that* $\mathbf{h}^k := \widehat{\Delta}(\boldsymbol{\theta}^k) \nabla_{\boldsymbol{\theta}} \widehat{f}(\boldsymbol{\theta}^k)$ *(*$\widehat{\Delta}(\boldsymbol{\theta}^k)$ *is given by* (8)*), it follows*

$$\|\mathbf{g}^k - \mathbf{h}^k\| \le C_3 (2 + \gamma) \omega^{1/3} L^3 \sqrt{m \log m \log(K/\delta)} + C_4 \omega^{4/3} L^4 \sqrt{m \log m} + C_5 \omega^2 L^4 m, \qquad (12)$$

*where* $\mathbf{g}^k$ *is the stochastic semi-gradient in neural adaptive TD, and*

$$\|\mathbf{g}^k\| \le (2 + \gamma) C_7 \sqrt{m \log(K/\delta)} \qquad (13)$$

*holds with probability at least* $1 - 2\delta - 3L^2 \exp(-C_6 m \omega^{2/3} L)$ *over the randomness of the initial point, where* $\{C_i > 0\}_{i=1,\ldots,7}$ *and* $C > 0$ *are universal constants.*

Lemma 1 has been proved by [Lemma 6.1, [53]] and [Lemma 4.4, [8]]. The bounds (10) and (11) are only dependent on the structure of the neural networks. Our theory is built on Lemma 1. Without the ReLU activation, Lemma 1 cannot be guaranteed to hold, and thus we only consider deep ReLU networks in this paper. Indeed, due to the property of deep ReLU networks, there has been a line of theoretical research on deep ReLU networks, see e.g. [54, 18, 40, 9, 55, 53]. From Lemma 1, with high probabilities, $v^k \le (2+\gamma)^2 C_7^2 m \log(K/\delta)k$ when $k \le K$. But this is the worst-case bound and may not always be achievable. From (4), we can see that $\mathbf{g}^k$ keeps the sparsity of $\nabla_{\boldsymbol{\theta}} f(\boldsymbol{\theta}^k; \phi(s_k, a_k))$ because

$$[f(\boldsymbol{\theta}^k; \phi(s_k, a_k)) - r(s_k, a_k) - \gamma f(\boldsymbol{\theta}^k; \phi(s_{k+1}, a_{k+1}))] \in \mathbb{R}.$$

Thus, the stochastic semi-gradient $\mathbf{g}^k$ can be very sparse, based on which the following bound is proposed.

**Condition 1** *In neural adaptive temporal difference learning, the following bound holds*

$$v^k \le C_0 [m \log(K/\delta)k]^\alpha, \ 0 < \alpha \le 1, \ 1 \le k \le K. \qquad (14)$$

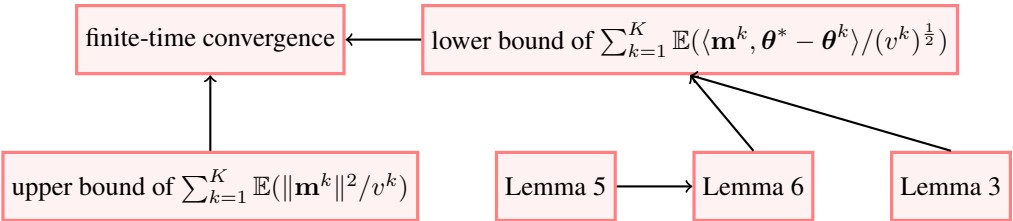

Figure 1: The roadmap of the proof.

We can see that $\alpha = 1$ directly holds for (14) due to the bound (13); while $0 < \alpha < 1$ indicates the stochastic semi-gradients decay fast. In applications, the sparse gradients can always obey such a condition. We stress that such assumption is standard in the analysis of the Adam-type algorithms; see, e.g., references [32, 20, 39, 11, 35, 45].

Now, we turn to the neural approximation case. As mentioned above, the convergence analysis uses the approximate stationary point $\theta^*$. One core part of our proofs is investigating $\mathbf{h}$ versus the iteration. Due to the Markovian observation, $s_k, a_k, s_{k+1}, a_{k+1}$ may miss visiting several states in a single sampling, i.e., choosing some states and actions with probability 0. That indicates $\mathbb{E}[\mathbf{h}^k]$ may be a biased expectation of $\mathbf{h}$. To get controlled difference bound, we consider $\mathbb{E}(\overline{\mathbf{h}}(\theta^{k-T}) \mid \chi^k)$ and $\mathbf{h}(\theta^{k-T})$, where

$$\overline{\mathbf{h}}(\theta^{k-T}) := \hat{\Delta}(s_k, a_k, s_{k+1}, a_{k+1}; \theta^{k-T}) \nabla_{\theta} \hat{f}(\theta^{k-T}; \phi(s_k, a_k)).$$

Although with biased samples, the Markov property means $\mathcal{P}(s_k = s \mid s_{k-T} = s', a_{k-T} = a')$ is sufficiently close to $\mu(s)$ when $T$ is large. Such a technique then yields following lemma.

**Lemma 2** *Assume $(\theta^k)_{0 \leq k \leq K}$ is generated by neural adaptive TD, and condition* (9) *holds. Given an integer $1 \leq T \leq K$, as $k \geq T$, we have*

$$\left\| \mathbb{E}(\overline{\mathbf{h}}(\theta^{k-T}) \mid \chi^k) - \mathbf{h}(\theta^{k-T}) \right\| \leq (2 + \gamma) C_3 C_4 \bar{\kappa} \sqrt{m \log \frac{K}{\delta}} \rho^T$$

*with probability at least $1 - 2\delta - 3L^2 \exp(-C_6 m \omega^{2/3} L)$ over the randomness of the initial point.*

The factor $\rho^T$ in the upper bound in Lemma 2 comes from the Markovian noise. The lemma tells that when $T$ is large enough, the difference between $\mathbb{E}\overline{\mathbf{h}}(\theta^{k-T})$ and $\mathbf{h}(\theta^{k-T})$ will be very small.

The lower bound directly holds based on the definitions above.

**Lemma 3** *For $\forall \theta \in \mathbf{V}$, it follows that*

$$\langle \mathbf{h}(\theta), \theta - \theta^* \rangle \geq (1 - \gamma) \mathbb{E}(\hat{f}(\theta; s, a) - \hat{f}(\theta^*; s, a))^2. \tag{15}$$

## 3  Main Results

This section contains the finite-time analysis of adaptive TD using deep neural network approximations. The descriptions of convergence results are dependent on the notion of the approximate stationary point $\theta^*$.

### 3.1  Roadmap

We first provide a high level explanation of the proofs and how the technical lemmas work in the proofs. Our proofs begin with a traditional idea of algorithmic convergence analysis, i.e., bounding the gap $(\|\theta^* - \theta^k\|^2)_{k \geq 0}$ versus the iteration. It is easy to have the following estimate

$$2\eta \langle \mathbf{m}^k, \theta^* - \theta^k \rangle / (v^k)^{\frac{1}{2}} \leq \|\theta^* - \theta^k\|^2 - \|\theta^* - \theta^{k+1}\|^2 + \eta^2 \|\mathbf{m}^k\|^2 / v^k.$$

Summing this inequality from $k = 1$ to $K$ and taking expectation, we are led to

$$2\eta \sum_{k=1}^{K} \mathbb{E}(\langle \mathbf{m}^k, \boldsymbol{\theta}^* - \boldsymbol{\theta}^k \rangle / (v^k)^{\frac{1}{2}}) \leq \mathbb{E}\|\boldsymbol{\theta}^* - \boldsymbol{\theta}^1\|^2 + \eta^2 \sum_{k=1}^{K} \mathbb{E}(\|\mathbf{m}^k\|^2 / v^k). \tag{16}$$

Notice that (16) is very similar to the recursion of the well-known SGD. Recalling the finite-time convergence analysis of SGD, we then get the big picture for the proof of our algorithms, which contains two major steps and can be presented by Figure 1.

- **Step 1**. The first step is quite straightforward, i.e., bounding the summation $\sum_{k=1}^{K} \mathbb{E}(\|\mathbf{m}^k\|^2 / v^k)$ with high probability. This step is proved by Lemma 4 in the supplementary materials.

- **Step 2**. The second step is to establish a lower bound of $\sum_{k=1}^{K} \mathbb{E}(\langle \mathbf{m}^k, \boldsymbol{\theta}^* - \boldsymbol{\theta}^k \rangle / (v^k)^{\frac{1}{2}})$. However, it is much more complicated than the first step. To this end, we need to exploit the relation between $\mathbb{E}\langle \mathbf{m}^k, \boldsymbol{\theta}^* - \boldsymbol{\theta}^k \rangle / (v^k)^{\frac{1}{2}}$ and $\mathbb{E}\langle \mathbf{h}(\boldsymbol{\theta}^k), \boldsymbol{\theta}^* - \boldsymbol{\theta}^k \rangle / (v^{k-1})^{\frac{1}{2}}$ (proved by Lemma 6 in the supplementary materials). Nevertheless, we cannot directly get Lemma 6; we have to develop a technical lemma that connects $\mathbb{E}\langle \boldsymbol{\theta}^k - \boldsymbol{\theta}^*, \mathbf{h}(\boldsymbol{\theta}^k) \rangle / (v^{k-1})^{\frac{1}{2}}$ with $\mathbb{E}\langle \boldsymbol{\theta}^* - \boldsymbol{\theta}^k, \mathbf{g}^k \rangle / (v^{k-1})^{\frac{1}{2}}$ (proved by Lemma 5 in the supplementary materials). Further with the connection between $\mathbb{E}\langle \mathbf{h}(\boldsymbol{\theta}^k), \boldsymbol{\theta}^k - \boldsymbol{\theta}^* \rangle$ and $\mathbb{E}(\hat{f}(\boldsymbol{\theta}^k; s, a) - \hat{f}(\boldsymbol{\theta}^*; s, a))^2$ (Lemma 3), we then prove the lower bound.

### 3.2 Finite-Time Analysis of Neural Adaptive TD

Now, we are prepared to present the convergence of neural adaptive TD.

**Theorem 1** *Suppose* $(\boldsymbol{\theta}^k)_{k \geq 0}$ *is generated by neural adaptive TD under the Markovian observation, and condition* (9) *holds, and Assumptions 1, 2, and condition* (14) *hold. Given the integer* $T \in \mathbb{Z}^+$, $\eta > 0, v_0 \geq \varpi > 0, 0 \leq \beta < 1$, *for* $K \geq 2^{\frac{2}{2-\alpha}}T$, *we have*

$$\min_{1 \leq k \leq K} \mathbb{E}(\hat{f}(\boldsymbol{\theta}^k; s, a) - \hat{f}(\boldsymbol{\theta}^*; s, a))^2 \leq \frac{c_1(m, \eta, \alpha, T)}{K^{1-\alpha/2}} + \frac{c_2(m, \eta, \omega, \alpha, T, K)}{K^{1-\alpha/2}} \\ + c_3(m, \omega, \alpha, K)\rho^T K^{\frac{\alpha}{2}} + c_4(m, \omega, K) \tag{17}$$

*with probability at least* $1 - 2\delta - 3L^2 \exp(-C_6 m\omega^{2/3}L)$ *over the randomness of the initial point, where*

$$
\begin{aligned}
c_1(m, \eta, \alpha, T) &= \widetilde{\mathcal{O}}([m^{\frac{\alpha}{2}}T^2\eta^2 + m^{\frac{\alpha}{2}}\eta]\log K), \\
c_2(m, \eta, \omega, \alpha, T, K) &= \mathcal{O}(\omega m^{\frac{1+\alpha}{2}}\log K + m^{\frac{1+\alpha}{2}}\omega^2 T + \omega\sqrt{m}/(K-T)), \\
c_3(m, \omega, \alpha, K) &= \mathcal{O}([\omega m^{\frac{\alpha+1}{2}}]\log K), \\
c_4(m, \omega, K) &= \widetilde{\mathcal{O}}([\omega^{\frac{4}{3}}\sqrt{m} + \omega^{\frac{7}{3}}\sqrt{m} + \omega^3 m]\log K),
\end{aligned} \tag{18}
$$

*and their details are given by* (29) *in supplementary materials.*

If we set the radius as $\omega = \Theta(m^{-1/2})$ [4], and $T = \frac{\ln(mK)}{\ln \frac{1}{\rho}}$, and $\eta = m^{-1/2}$, with (18), the right-hand side of (17) is in the order of

$$\widetilde{\mathcal{O}}\left(1/(K^{1-\alpha/2}\ln^2 \frac{1}{\rho}) + 1/(K^{1-\alpha/2}\ln \frac{1}{\rho}) + m^{-\frac{1}{6}}\right).$$

Thus, with high probability to achieve the $\epsilon$-accuracy for $\min_{1 \leq k \leq K} \mathbb{E}_\pi(\hat{f}(\boldsymbol{\theta}^k; s, a) - \hat{f}(\boldsymbol{\theta}^*; s, a))^2$, we need that

$$m = \tilde{\Theta}(1/\epsilon^6), \ 1/(K^{1-\alpha/2}\ln \frac{1}{\rho}) = \widetilde{\mathcal{O}}(\epsilon).$$

Thus, we get the worst-case iteration complexity of $K$ as follows

$$\widetilde{\mathcal{O}}\left(\frac{1}{\epsilon^{\frac{2}{2-\alpha}} \cdot \ln^{\frac{2}{2-\alpha}}(\frac{1}{\rho})}\right). \tag{19}$$

---

[4]Such a choice of the radius in neural TD is also used in [53].

The worse case is $\alpha = 1$, in which case we get the complexity as $\widetilde{\mathcal{O}}(1/\epsilon^2)$. Note that even for SGD without strong convexity, the optimal complexity is $\mathcal{O}(1/\epsilon^2)$ [Theorem 4, [17]]. Thus, our result is nearly optimal (just with an additional logarithmic factor that barely hurting the rate). We can see that when $0 \le \alpha < 1$, the worst iteration complexity of $K$ is smaller than $\widetilde{\mathcal{O}}\left(\frac{1}{\epsilon^2 \ln^2(\frac{1}{\rho})}\right)$ when $\epsilon \ll 1/[\ln(\frac{1}{\rho})]$, which indicates a *faster speed than the neural TD*. Due to that $\epsilon$ is the desired error and thus can be very small, the acceleration always happens for a small $\alpha$. Note that $\alpha = 1$ directly holds with any extra assumption. In this case, the iteration complexity of the adaptive neural TD matches existing neural TD [53] in the case of DNN approximation.

It is worth mentioning that compared with the convergence rate of neural TD with ReLU deep networks, we used the same searching radius and network width as presented by the authors of [53]. In applications, the sparse stochastic semi-gradients always yield the fast decaying condition. In other words, with sparse stochastic semi-gradients, the neural adaptive TD also accelerates the vanilla TD even in the ReLU network approximation case. Such a phenomenon resonates with the existing acceleration results of adaptive methods for stochastic optimization: "for sparse data, the adaptive methods are likely to perform better than non-adaptive methods" [20].

The scheme of neural TD uses a projection to the set $\mathbf{V}$ such that all iterates are constrained in a small neighborhood around the initialization (also called NTK regime). Such a procedure is to use the property of deep ReLU networks [7, 53]. Thus, the searching radius is small, which is reasonable because in DNN training, the parameters usually only change in a very small range in the overparameterized regime.

**Proposition 1** *Assume that conditions of Theorem 1 hold, and Condition 1 holds with $0 \le \alpha < 1, \omega = \Theta(m^{-1/2}), \eta = m^{-\frac{1}{2}}, m = \widetilde{\Theta}(1/\epsilon^6), \epsilon \ll 1/[\ln(\frac{1}{\rho})]$, the neural adaptive TD enjoys a faster finite-time convergence rate than the neural TD in the ReLU DNN approximation of function values with probability at least $1 - 2\delta - 3L^2 \exp(-C_6 m \omega^{2/3} L)$ over the randomness of the initial point.*

In the following, we present a proposition that characterizes the difference between our established results and the optimal action-value function.

**Proposition 2** *Assume conditions of Proposition 1 hold, with probability at least $1 - 2\delta - 3L^2 \exp(-C_6 m \omega^{2/3} L)$ over the randomness of the initial point, we have*

$$\min_{1 \le k \le K} \mathbb{E}\big(f(\boldsymbol{\theta}^k; s, a) - \mathbf{Q}^*(s, a)\big)^2$$

$$= \widetilde{\mathcal{O}}\left(\frac{\mathbb{E}[(\Pi_{\mathcal{F}_{\mathbf{V},m}}(\mathbf{Q}^*(s,a)) - \mathbf{Q}^*(s,a))^2]}{1 - \gamma} + \frac{1}{\epsilon^{\frac{2}{2-\alpha}} \cdot \ln^{\frac{2}{2-\alpha}}(\frac{1}{\rho})}\right),$$

*where $\Pi_{\mathcal{F}_{\mathbf{V},m}}(\mathbf{Q}^*(s,a))$ is projection of $\mathbf{Q}^*(s,a)$ to the linear function family $\mathcal{F}_{\mathbf{V},m}$, that is, $\Pi_{\mathcal{F}_{\mathbf{V},m}}(\mathbf{Q}^*(s,a)) := f(\boldsymbol{\theta}^{init}; \phi(s,a)) + \langle \nabla_{\boldsymbol{\theta}} f(\boldsymbol{\theta}^{init}; \phi(s,a)), \boldsymbol{\theta}^\dagger - \boldsymbol{\theta}^{init}\rangle$ with $\boldsymbol{\theta}^\dagger \in \arg\min_{\boldsymbol{\theta} \in \mathbf{V}} \left\{\left\|f(\boldsymbol{\theta}^{init}; \phi(s,a)) + \langle \nabla_{\boldsymbol{\theta}} f(\boldsymbol{\theta}^{init}; \phi(s,a)), \boldsymbol{\theta} - \boldsymbol{\theta}^{init}\rangle - \mathbf{Q}^*(s,a)\right\|\right\}$.*

Proposition 2 indicates that the algorithm can find the optimal action-value function $\mathbf{Q}^*(s, a)$ provided that the function family $\mathcal{F}_{\mathbf{V},m}$ contains $\mathbf{Q}^*(s, a)$.

## 4 Conclusions

This paper studies the finite-time convergence analyses of temporal difference learning with adaptive learning rates and momentum approximated by deep ReLU neural networks using the Markovian samplings. Our established theoretical results show that the neural adaptive temporal difference learning is convergent when the neural network is sufficiently wide. Our work shows convergence results and establishes the theoretical advantages of the adaptive algorithms for neural network approximation cases, i.e., adaptive schemes achieve better rates than neural temporal difference learning when the stochastic semi-gradients decay fast. There are numerous avenues for future works, including 1) Can we extend the multiple ReLU active functions to others, e.g., sigmoid or more general functions? 2) Can we establish the finite-time convergence of adaptive temporal difference

learning with other kinds of neural network approximation, e.g., recurrent neural networks and graph neural networks? 3) Can we get a relaxed bound for the radius of the searching area?

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
