# Supplementary materials for

## *Finite-Time Analysis of Adaptive Temporal Difference Learning with Deep Neural Networks*

## Nomenclature

$L, m$    Length and width of the deep ReLU.

$\mathcal{A}$    The action space.

$\beta$    Momentum hyper-paramter.

$\eta$    Hyper-paramter in the adaptive stepsize.

$\mathbf{h}(\boldsymbol{\theta})$    Used in Definition 1 and mathematically defined as $\mathbf{h}(\boldsymbol{\theta}) := \mathbb{E}\left[\widehat{\Delta}(\boldsymbol{\theta}) \nabla_{\boldsymbol{\theta}} \widehat{f}(\boldsymbol{\theta}; \phi(s,a))\right]$.

$\mathcal{F}_{\mathbf{V},m}, \hat{f}$    The collection of all local linearization of $f(\boldsymbol{\theta}; \phi(s,a))$ at the initial point $\boldsymbol{\theta}^{\text{init}}$, $\hat{f}$ is element of $\mathcal{F}_{\mathbf{V},m}$

$\overline{\mathbf{h}}(\boldsymbol{\theta}^{k-T})$    Technical item defined as $\overline{\mathbf{h}}(\boldsymbol{\theta}^{k-T}) := \widehat{\Delta}(s_k, a_k, s_{k+1}, a_{k+1}; \boldsymbol{\theta}^{k-T}) \nabla_{\boldsymbol{\theta}} \widehat{f}(\boldsymbol{\theta}^{k-T}; \phi(s_k, a_k))$.

$\mathbf{Q}_\pi$    Action-value function with policy $\pi$.

$\mathcal{T}_\pi$    Bellman operator associated with $\pi$.

$\boldsymbol{\theta}^{\text{init}}$    The initial point.

$f(\boldsymbol{\theta}; \mathbf{x})$    L-hidden-layer ReLU neural network defined as $f(\boldsymbol{\theta}; \mathbf{x}) = \sqrt{m}\mathbf{W}_L \sigma(\mathbf{W}_{L-1} \cdots \sigma(\mathbf{W}_1 \mathbf{x}) \cdots)$, where $\mathbf{x} \in \mathbb{R}^d$ is the input data, $\mathbf{W}_1 \in \mathbb{R}^{m \times d}$, $\mathbf{W}_L \in \mathbb{R}^{1 \times m}$ and $\mathbf{W}_l \in \mathbb{R}^{m \times m}$ for $l = 2, \ldots, L-1$, $\boldsymbol{\theta} := [\mathbf{W}_1, \ldots, \mathbf{W}_L]$ denotes all the weights.

$\mathbb{E}[\cdot]$    Expectation with respect to the underlying probability space *without* stochasticity of the initial point.

$\gamma$    The discount factor.

$\widehat{\Delta}$    Temporal difference error defined by (8).

$\mu$    Stationary distribution of the states.

$\omega$    Radius.

$\overline{\mathbf{g}}(\boldsymbol{\theta}; s_k, a_k, s_{k+1}, a_{k+1})$    Semi-gradient sampling operator denoted by (4).

$\mathcal{P}_a$    The transition matrix associated with action $a$.

$\phi(s,a) : \mathcal{S} \times \mathcal{A} \to \mathbb{R}^d$    State-action feature mapping.

$\pi, \pi(s,a)$    Policy, the probability to choose action $a$ when the current state is $s$.

$\mathbf{Proj}_{\mathbf{V}}(\mathbf{x})$    Projection of $\mathbf{x}$ onto set $\mathbf{V}$.

$\mathcal{S}$    The state space.

$\sigma(\cdot)$    ReLU activation function.

$\boldsymbol{\theta}^*$    Approximate stationary point (Definition 1).

$\boldsymbol{\theta}^k, \mathbf{m}^k, v^k$    The value, momentum and sum of past stochastic semi-gradients' norms in the $k$th iteration of adaptive TD with DNN.

$r(s,a)$    The reward with pair $(s,a)$.

$\mathbf{B}(\boldsymbol{\theta}, \omega)$    The ball centred at $\boldsymbol{\theta}$ with radius $\omega$.

$\mathbf{g}^k$    Stoch semi-gradient in $k$th iteration.

$\mathbf{V}_\pi$    Value function associated with $\pi$.

# A Other Technical Lemmas

In the proofs, we use three shorthand notations for simplifications. Those three notations are all related to the iteration $k$. Assume $(\mathbf{m}^k)_{k\geq 0}$, $(\boldsymbol{\theta}^k)_{k\geq 0}$, $(v^k)_{k\geq 0}$ are all generated by the neural adaptive TD. We denote

$$
\begin{aligned}
\Xi_k &:= \mathbb{E}\left(\|\mathbf{m}^k\|^2/(v^k)\right), \\
\Upsilon_k &:= \mathbb{E}\left(\langle \boldsymbol{\theta}^k - \boldsymbol{\theta}^*, \mathbf{m}^k\rangle/(v^k)^{\frac{1}{2}}\right),
\end{aligned}
$$

$$
\begin{aligned}
\Re_k &:= (1-\beta)(1+\gamma)C_3 C_4 \sqrt{m\log(K/\delta)} \sum_{h=1}^{T} \Xi_{k-h} \\
&\quad + (1-\beta)(1+\gamma)C_3 C_4 \sqrt{m\log(K/\delta)}\frac{(1-\beta)\omega^2 LT}{\varpi} \\
&\quad + \eta\beta\Xi_k + \frac{(1-\beta)(2+\gamma)C_3 C_4 \omega\bar{\kappa}\sqrt{Lm\log\frac{K}{\delta}}}{\sqrt{\varpi}}\rho^T \\
&\quad + \omega(2+\gamma)C_7\sqrt{Lm\log(K/\delta)}\left[\frac{1}{(v^{k-1})^{\frac{1}{2}}} - \frac{1}{(v^k)^{\frac{1}{2}}}\right] \\
&\quad + \frac{\omega\sqrt{L}(1-\beta)}{(v^k)^{\frac{1}{2}}}\left(C_3(2+\gamma)\omega^{1/3}L^3\sqrt{m\log m\log(K/\delta)}\right. \\
&\quad \left. + C_4\omega^{4/3}L^4\sqrt{m\log m} + C_5\omega^2 L^4 m\right).
\end{aligned} \tag{20}
$$

The technical lemmas are all described using the notations given above.

**Lemma 4** *Let $(\Xi_k)_{k\geq 0}$ be defined in (20) and $v^1 \geq \varpi > 0$, then we have*

$$
\sum_{k=1}^{K} \Xi_k \leq \sum_{j=1}^{K-1} \|\mathbf{g}^j\|^2/v^j.
$$

*Further, if condition (9) holds, we then get*

$$
\sum_{k=1}^{K} \Xi_k \leq \log\left[\frac{(K-1)(2+\gamma)^2 C_7^2 m\log(K/\delta)}{\varpi}\right].
$$

*with probability at least $1 - 2\delta - 3L^2\exp(-C_6 m\omega^{2/3}L)$ over the randomness of the initial point.*

**Lemma 5** *Assume condition (9) holds, given $T \in \mathbb{Z}^+$, with probability at least $1 - 2\delta - 3L^2\exp(-C_6 m\omega^{2/3}L)$, we have*

$$
\begin{aligned}
\mathbb{E}\left[\langle \boldsymbol{\theta}^k - \boldsymbol{\theta}^*, \mathbf{g}^k\rangle/(v^{k-1})^{\frac{1}{2}}\right] &\leq \mathbb{E}\left[\langle \boldsymbol{\theta}^k - \boldsymbol{\theta}^*, \mathbf{h}(\boldsymbol{\theta}^k)\rangle/(v^{k-1})^{\frac{1}{2}}\right] \\
&\quad + \frac{\omega\sqrt{L}}{(v^k)^{\frac{1}{2}}}\left(C_3(2+\gamma)\omega^{1/3}L^3\sqrt{m\log m\log(K/\delta)} + C_4\omega^{4/3}L^4\sqrt{m\log m} + C_5\omega^2 L^4 m\right) \\
&\quad + \frac{(2+\gamma)C_3 C_4 \omega\bar{\kappa}\sqrt{Lm\log\frac{K}{\delta}}\rho^T}{\sqrt{\varpi}} + (1+\gamma)C_3 C_4\sqrt{m\log(K/\delta)}\left(\sum_{h=1}^{T}\frac{\mathbb{E}\|\mathbf{m}^{k-h}\|^2}{v^{k-h}} + \frac{\omega^2 LT}{\varpi}\right).
\end{aligned}
$$

**Lemma 6** *Let $(\Upsilon_k)_{k\geq 0}$ and $(\Re_k)_{k\geq 0}$ be defined in (20), then the following result holds for neural adaptive TD*

$$
\Upsilon_k + (1-\beta)\mathbb{E}\left(\langle \boldsymbol{\theta}^k - \boldsymbol{\theta}^*, \mathbf{h}(\boldsymbol{\theta}^k)\rangle/(v^{k-1})^{\frac{1}{2}}\right) \leq \beta\Upsilon_{k-1} + \Re_k. \tag{21}
$$

# B  Proof of Theorem 1

The bounds in the proof are all with probability at least $1 - 2\delta - 3L^2 \exp(-C_6 m \omega^{2/3} L)$. Given $K \in \mathbb{Z}^+$, summing $k = 1$ to $K$ of (21) gives us

$$(1 - \beta) \sum_{k=T+1}^{K} \mathbb{E}\Big( \langle \boldsymbol{\theta}^k - \boldsymbol{\theta}^*, \mathbf{h}(\boldsymbol{\theta}^k) \rangle / (v^{k-1})^{\frac{1}{2}} \Big)$$

$$\leq -\Upsilon_K + (1 - \beta) \sum_{k=T}^{K-1} (-\Upsilon_k) + \sum_{k=T+1}^{K} \Re_k \tag{22}$$

$$\leq (1 - \beta) \sum_{k=T}^{K-1} (-\Upsilon_k) + \sum_{k=T+1}^{K} \Re_k + \frac{\omega(2 + \gamma)C_7 \sqrt{m \log(K/\delta)}}{(v^K)^{\frac{1}{2}}},$$

where we used the fact that $\mathbf{m}^k \leq (2 + \gamma)C_7 \sqrt{m \log(K/\delta)}$ when $k \leq K$. The convex projection is contractive,

$$\|\boldsymbol{\theta}^* - \boldsymbol{\theta}^{k+1}\|^2 \leq \|\boldsymbol{\theta}^* - \boldsymbol{\theta}^k - \eta \mathbf{m}^k / (v^k)^{\frac{1}{2}}\|^2$$

$$\leq \|\boldsymbol{\theta}^* - \boldsymbol{\theta}^k\|^2 + 2\eta \langle \mathbf{m}^k, \boldsymbol{\theta}^k - \boldsymbol{\theta}^* \rangle / (v^k)^{\frac{1}{2}} + \eta^2 \|\mathbf{m}^k\|^2 / v^k.$$

Taking the total condition expectation gives us

$$\mathbb{E}\|\boldsymbol{\theta}^* - \boldsymbol{\theta}^{k+1}\|^2 \leq \mathbb{E}\|\boldsymbol{\theta}^* - \boldsymbol{\theta}^k\|^2 + 2\eta \Upsilon_k + \eta^2 \Xi_k,$$

which directly indicates the following inequality

$$\sum_{k=T}^{K-1} -\Upsilon_k \leq \frac{\mathbb{E}\|\boldsymbol{\theta}^* - \boldsymbol{\theta}^T\|^2}{2\eta} + \frac{\eta}{2} \sum_{k=T}^{K-1} \Xi_k.$$

With (22), we can derive

$$\sum_{k=T+1}^{K} \mathbb{E}\Big( \langle \boldsymbol{\theta}^k - \boldsymbol{\theta}^*, \mathbf{h}(\boldsymbol{\theta}^k) \rangle / (v^{k-1})^{\frac{1}{2}} \Big)$$

$$\leq \sum_{k=T}^{K-1} (-\Upsilon_k) + \frac{1}{1 - \beta} \sum_{k=T+1}^{K} \Re_k + \omega(2 + \gamma)C_7 \sqrt{m \log(K/\delta)} / [(1 - \beta)(\varpi)^{\frac{1}{2}}]$$

$$\leq \frac{\mathbb{E}\|\boldsymbol{\theta}^* - \boldsymbol{\theta}^T\|^2}{\eta} + \eta \sum_{k=T}^{K-1} \Xi_k + \frac{1}{1 - \beta} \sum_{k=T+1}^{K} \Re_k + \omega(2 + \gamma)C_7 \sqrt{m \log(K/\delta)} / [(1 - \beta)(v^K)^{\frac{1}{2}}].$$

$$\tag{23}$$

We use the following shorthand notations

$$\aleph_0 = (1 - \beta)(1 + \gamma)C_3 C_4 \sqrt{m \log(K/\delta)} \frac{(1 - \beta)\omega^2 LT}{\varpi},$$

$$\aleph_1 := \frac{(2 + \gamma)C_3 C_4 \omega \bar{\kappa} \sqrt{Lm \log \frac{K}{\delta}}}{\sqrt{\varpi}},$$

$$\aleph_2 := \omega(2 + \gamma)C_7 \sqrt{Lm \log(K/\delta)},$$

$$\aleph_3 := \omega \sqrt{L} (C_3 (2 + \gamma)\omega^{1/3} L^3 \sqrt{m \log m \log(K/\delta)}$$

$$+ C_4 \omega^{4/3} L^4 \sqrt{m \log m} + C_5 \omega^2 L^4 m).$$

Using Lemma 7 and Lemma 4, we have the following bound

$$\eta \sum_{k=T}^{K-1} \Xi_k + \frac{1}{1-\beta} \sum_{k=T+1}^{K} \Re_k$$

$$\leq (1+\gamma)C_3 C_4 \eta \sqrt{m \log(K/\delta)} \sum_{k=T+1}^{K} \sum_{j=1}^{T} \Xi_{k-j} + \eta \sum_{k=T}^{K-1} \Xi_k + \frac{\eta\beta}{1-\beta} \sum_{k=T+1}^{K} \Xi_k + \frac{\aleph_2}{(v^K)^{1/2}}$$

$$+ \aleph_1 \rho^T (K-T) + \aleph_0(K-T) + \sum_{k=T}^{K} \frac{\aleph_3}{(v^{k-1})^{\frac{1}{2}}}$$

$$\leq \left( \eta + (1+\gamma)C_3 C_4 \eta \sqrt{m \log(K/\delta)} T^2 + \frac{\eta\beta}{1-\beta} \right) \times \sum_{k=1}^{K} \Xi_k$$

$$+ \frac{\aleph_2}{(v^K)^{1/2}} + \aleph_1 \rho^T (K-T) + \aleph_0(K-T) + \sum_{k=T}^{K} \frac{\aleph_3}{(v^{k-1})^{\frac{1}{2}}}.$$

Further with Lemma 4, the upper bound of right side is bounded by

$$\left( \eta + (1+\gamma)C_3 C_4 \eta \sqrt{m \log(K/\delta)} T^2 + \frac{\eta\beta}{1-\beta} \right) \times \log \left[ \frac{(K-1)(2+\gamma)^2 C_7^2 m \log(K/\delta)}{\varpi} \right]$$

$$+ \frac{\aleph_2}{(v^K)^{1/2}} + \aleph_1 \rho^T (K-T) + \aleph_0(K-T) + \sum_{k=T}^{K} \frac{\aleph_3}{(v^{k-1})^{\frac{1}{2}}}.$$

$$(24)$$

On the other hand, we have

$$\sum_{k=T}^{K} \mathbb{E} \langle \boldsymbol{\theta}^k - \boldsymbol{\theta}^*, \mathbf{h}(\boldsymbol{\theta}^k) \rangle / (v^{k-1})^{\frac{1}{2}}$$

$$\geq \sum_{k=T}^{K} \frac{(1-\gamma)\mathbb{E}\big( \hat{f}(\boldsymbol{\theta}^k; s, a) - \hat{f}(\boldsymbol{\theta}^*; s, a) \big)^2}{(v^{k-1})^{\frac{1}{2}}} \qquad (25)$$

$$\geq \Big[ \sum_{k=T}^{K} \frac{(1-\gamma)}{(v^{k-1})^{\frac{1}{2}}} \Big] \cdot \min_{T \leq k \leq K} \mathbb{E}\big( \hat{f}(\boldsymbol{\theta}^k; s, a) - \hat{f}(\boldsymbol{\theta}^*; s, a) \big)^2.$$

Thus, we can get

$$\min_{T \leq k \leq K} \mathbb{E}\big( \hat{f}(\boldsymbol{\theta}^k; s, a) - \hat{f}(\boldsymbol{\theta}^*; s, a) \big)^2$$

$$\leq \left( \eta + (1+\gamma)C_3 C_4 \eta \sqrt{m \log(K/\delta)} T^2 + \frac{\eta\beta}{1-\beta} \right)$$

$$\times \log \left[ \frac{(K-1)(2+\gamma)^2 C_7^2 m \log(K/\delta)}{\varpi} \right] \Big/ \Big[ \sum_{k=T}^{K} \frac{(1-\gamma)}{(v^{k-1})^{\frac{1}{2}}} \Big] \qquad (26)$$

$$+ \frac{\frac{(1+\beta)\aleph_2}{(v^K)^{1/2}} + (\aleph_1 \rho^T + \aleph_0)(K-T) + \sum_{k=T}^{K} \frac{\aleph_3}{(v^{k-1})^{\frac{1}{2}}} + \frac{L\omega^2}{\eta}}{\big( \sum_{k=T}^{K} \frac{(1-\gamma)}{(v^{k-1})^{\frac{1}{2}}} \big)}.$$

Notice that $(v^k)_{k \geq 0}$ is increasing, $\sum_{k=T}^{K} \frac{(1-\gamma)}{(v^{k-1})^{\frac{1}{2}}} \geq \frac{(K-T)(1-\gamma)}{(v^{K-1})^{\frac{1}{2}}}$, and thus

$$\Big[ \frac{(1+\beta)\aleph_2}{(v^K)^{1/2}} \Big] \Big/ \Big[ \sum_{k=T}^{K} \frac{(1-\gamma)}{(v^{k-1})^{\frac{1}{2}}} \Big] \leq \frac{(1+\beta)\aleph_2}{(K-T)(1-\gamma)} \frac{(v^{K-1})^{\frac{1}{2}}}{(v^K)^{\frac{1}{2}}} \leq \frac{(1+\beta)\aleph_2}{(K-T)(1-\gamma)}. \qquad (27)$$

On the other hand, from Lemma 1, with high probabilities, $v^k \leq (2+\gamma)^2 C_7^2 m \log(K/\delta)k$ when $k \leq K$, and then we can get

$$\sum_{k=T}^{K} 1/(v^{k-1})^{\frac{1}{2}} \geq \sum_{k=T}^{K} \frac{1}{C_0(m\log(K/\delta)k)^{\alpha/2}} \geq \frac{2(K^{1-\alpha/2} - T^{1-\alpha/2})}{\alpha C_0(m\log(K/\delta))^{\alpha/2}} \geq \frac{K^{1-\alpha/2}}{\alpha C_0(m\log(K/\delta))^{\alpha/2}},$$
(28)

where we used $K \geq 2^{\frac{2}{2-\alpha}}T$ to get $2(K^{1-\alpha/2} - T^{1-\alpha/2}) \geq K^{1-\alpha/2}$. Combing (27), (28) and (26), we are led to

$$\min_{1 \leq k \leq K} \mathbb{E}\big(\hat{f}(\boldsymbol{\theta}^k; s, a) - \hat{f}(\boldsymbol{\theta}^*; s, a)\big)^2$$

$$\leq \min_{T \leq k \leq K} \mathbb{E}\big(\hat{f}(\boldsymbol{\theta}^k; s, a) - \hat{f}(\boldsymbol{\theta}^*; s, a)\big)^2$$

$$\leq \left((1+\gamma)C_3 C_4 \eta \sqrt{m\log(K/\delta)}T^2 + \frac{\eta + \eta\beta}{(1-\gamma)(1-\beta)}\right) \times \log\left[\frac{(K-1)(2+\gamma)^2 C_7^2 m\log(K/\delta)}{\varpi}\right]$$

$$\times C_0(m\log(K/\delta))^{\alpha/2}/K^{1-\alpha/2} + \frac{\omega(2+\gamma)C_0 C_7 L[m\log(K/\delta)]^{\frac{1+\alpha}{2}}}{(1-\gamma)(1-\beta)\sqrt{\varpi}}/K^{1-\alpha/2}$$

$$+ (1-\beta)^2(1+\gamma)C_0 C_3 C_4(m\log(K/\delta))^{\frac{\alpha+1}{2}}\frac{\omega^2 LT}{\varpi}/K^{1-\alpha/2}$$

$$+ \frac{(2+\gamma)C_0 C_3 C_7 \omega\bar{\kappa}\sqrt{Lm\log\frac{K}{\delta}}(m\log(K/\delta))^{\alpha/2}}{\sqrt{\varpi}(1-\gamma)}\rho^T K^{\alpha/2}$$

$$+ \frac{\omega\sqrt{L}(1-\beta)}{(1-\gamma)}(C_3(2+\gamma)\omega^{1/3}L^3\sqrt{m\log m\log(K/\delta)}$$

$$+ C_4\omega^{4/3}L^4\sqrt{m\log m} + C_5\omega^2 L^4 m) + \frac{\frac{L\omega^2}{\eta}C_0(m\log(K/\delta))^{\alpha/2}}{(1-\gamma)K^{1-\alpha/2}} + \frac{2(2+\gamma)C_7\omega\sqrt{Lm\log(K/\delta)}}{(K-T)(1-\gamma)}.$$

Letting

$$c_1(m, \eta, \alpha, T, K) := \left((1+\gamma)C_3 C_4 \eta \sqrt{m\log(K/\delta)}T^2 + \frac{\eta + \eta\beta}{(1-\gamma)(1-\beta)}\right)$$

$$\times \log\left[\frac{(K-1)(2+\gamma)^2 C_7^2 m\log(K/\delta)}{\varpi}\right]C_0(m\log(K/\delta))^{\frac{\alpha}{2}},$$

$$c_2(m, \eta, \omega, \alpha, T, K) := \frac{2\omega(2+\gamma)C_0 C_7 L[m\log(K/\delta)]^{\frac{1+\alpha}{2}}}{(1-\gamma)(1-\beta)\sqrt{\varpi}}$$

$$+ \frac{\frac{L\omega^2}{\eta}C_0(m\log(K/\delta))^{\alpha/2}}{1-\gamma} + \frac{2(2+\gamma)C_7\omega\sqrt{Lm\log(K/\delta)}}{(K-T)(1-\gamma)}$$
(29)

$$+ (1-\beta)^2(1+\gamma)C_0 C_3 C_4(m\log(K/\delta))^{\frac{\alpha+1}{2}}\frac{\omega^2 LT}{\varpi},$$

$$c_3(m, \omega, \alpha, K) := \frac{2(2+\gamma)C_0 C_3 C_7 \omega\bar{\kappa}\sqrt{Lm\log\frac{K}{\delta}}(m\log(K/\delta))^{\alpha/2}}{\sqrt{\varpi}(1-\gamma)},$$

$$c_4(m, \omega, K) := \frac{\omega\sqrt{L}(1-\beta)}{(1-\gamma)}\Big(C_3(2+\gamma)\omega^{1/3}L^3\sqrt{m\log m\log(K/\delta)}$$

$$+ C_4\omega^{4/3}L^4\sqrt{m\log m} + C_5\omega^2 L^4 m\Big),$$

which complete the proof.

## C Proof of Proposition 2

The proof is similar to the the proof of [Theorem 5.6,[53]] and is presented here for completeness. With the Cauchy's inequality,

$$\mathbb{E}\big(f(\boldsymbol{\theta}^k; s, a) - \mathbf{Q}^*(s, a)\big)^2 \leq 3\mathbb{E}\big(f(\boldsymbol{\theta}^k; s, a) - \hat{f}(\boldsymbol{\theta}^k; s, a)\big)^2$$
$$+ 3\mathbb{E}\big(\hat{f}(\boldsymbol{\theta}^k; s, a) - \hat{f}(\boldsymbol{\theta}^*; s, a)\big)^2 + 3\mathbb{E}\big(\hat{f}(\boldsymbol{\theta}^*; s, a) - \mathbf{Q}^*(s, a)\big)^2. \tag{30}$$

With (Theorems 5.3 and 5.4 in [8]] and $\omega = \Theta(m^{-1/2})$, we have

$$\mathbb{E}\big(f(\boldsymbol{\theta}^k; s, a) - \hat{f}(\boldsymbol{\theta}^k; s, a)\big)^2 = \widetilde{\mathcal{O}}(m^{-1/3})$$

with probability at least $1 - \delta$.

Notice that that $\hat{f}(\boldsymbol{\theta}^*; s, a)$ is the fixed point of $\Pi_{\mathcal{F}_{\mathbf{V},m}}\mathcal{T}_\pi(\cdot)$ and $\mathbf{Q}^*(s, a)$ is the fixed point of $\mathcal{T}_\pi(\cdot)$, respectively. For any $(s, a)$, we thus have

$$|\hat{f}(\boldsymbol{\theta}^*; s, a) - \mathbf{Q}^*(s, a)| = |\hat{f}(\boldsymbol{\theta}^*; s, a) - \Pi_{\mathcal{F}_{\mathbf{V},m}}\mathcal{T}_\pi(\mathbf{Q}^*(s, a)) + \Pi_{\mathcal{F}_{\mathbf{V},m}}\mathcal{T}_\pi(\mathbf{Q}^*(s, a)) - \mathbf{Q}^*(s, a)|$$

$$= |\mathbf{Proj}_{\mathcal{F}_{\mathbf{V},m}}\mathcal{T}_\pi(\hat{f}(\boldsymbol{\theta}^*; s, a)) - \Pi_{\mathcal{F}_{\mathbf{V},m}}\mathcal{T}_\pi(\mathbf{Q}^*(s, a)) + \Pi_{\mathcal{F}_{\mathbf{V},m}}\mathcal{T}_\pi(\mathbf{Q}^*(s, a)) - \mathbf{Q}^*(s, a)|$$

$$= |\mathbf{Proj}_{\mathcal{F}_{\mathbf{V},m}}\mathcal{T}_\pi(\hat{f}(\boldsymbol{\theta}^*; s, a)) - \Pi_{\mathcal{F}_{\mathbf{V},m}}\mathcal{T}_\pi(\mathbf{Q}^*(s, a)) + \Pi_{\mathcal{F}_{\mathbf{V},m}}(\mathbf{Q}^*(s, a)) - \mathbf{Q}^*(s, a)|$$

$$\leq \gamma|\hat{f}(\boldsymbol{\theta}^*; s, a) - \mathbf{Q}^*(s, a)| + |\Pi_{\mathcal{F}_{\mathbf{V},m}}(\mathbf{Q}^*(s, a)) - \mathbf{Q}^*(s, a)|,$$

where we used that fact that $\Pi_{\mathcal{F}_{\mathbf{V},m}}\mathcal{T}_\pi(\cdot)$ is $\gamma$-contract. Hence, we are led to

$$|\hat{f}(\boldsymbol{\theta}^*; s, a) - \mathbf{Q}^*(s, a)| \leq \frac{|\Pi_{\mathcal{F}_{\mathbf{V},m}}(\mathbf{Q}^*(s, a)) - \mathbf{Q}^*(s, a)|}{1 - \gamma}.$$

Turing back to (30),

$$\mathbb{E}\big(f(\boldsymbol{\theta}^k; s, a) - \mathbf{Q}^*(s, a)\big)^2$$

$$= \widetilde{\mathcal{O}}(m^{-1/3} + \mathbb{E}\big(\hat{f}(\boldsymbol{\theta}^k; s, a) - \hat{f}(\boldsymbol{\theta}^*; s, a)\big)^2 + \frac{\mathbb{E}[(\Pi_{\mathcal{F}_{\mathbf{V},m}}(\mathbf{Q}^*(s, a)) - \mathbf{Q}^*(s, a))^2]}{(1 - \gamma)^2}).$$

Note that $\mathbb{E}\big(\hat{f}(\boldsymbol{\theta}^k; s, a) - \hat{f}(\boldsymbol{\theta}^*; s, a)\big)^2$ has been bounded by Proposition 1, we then proved the result.

## D Proofs of Technical Lemmas

### D.1 Proof of Lemma 2

Given a fixed integer $T$, direct calculations give us

$$\mathbb{E}(\overline{\mathbf{h}}(\boldsymbol{\theta}^{k-T}; s_k, a_k, s_{k+1}, a_{k+1}) \mid \sigma^{k-T})$$

$$= \sum_{s,s'\in\mathcal{S}, a, a'\in\mathcal{A}} \mathcal{P}(s_k = s \mid s_{k-T}, a_{k-T})\mathcal{P}(a, s', a'|s)$$

$$\times \nabla_{\boldsymbol{\theta}}\widehat{f}\left(\boldsymbol{\theta}^{k-T}; \phi(s, a)\right)\widehat{\Delta}\left(\boldsymbol{\theta}^{k-T}; s, a, s', a'\right)$$

$$= \sum_{s,s'\in\mathcal{S}, a, a'\in\mathcal{A}} \mu(s)\mathcal{P}(a, s', a'|s)\nabla_{\boldsymbol{\theta}}\widehat{f}\left(\boldsymbol{\theta}^{k-T}; \phi(s, a)\right) \times \widehat{\Delta}\left(\boldsymbol{\theta}^{k-T}; s, a, s', a'\right) \tag{31}$$

$$+ \sum_{s,s'\in\mathcal{S}, a, a'\in\mathcal{A}} \mathcal{P}(a, s', a'|s)(\mathcal{P}(s_k = s \mid s_{k-T}, a_{k-T}) - \mu(s))\nabla_{\boldsymbol{\theta}}\widehat{f}\left(\boldsymbol{\theta}^{k-T}; \phi(s, a)\right)$$

$$\times \widehat{\Delta}\left(\boldsymbol{\theta}^{k-T}; s, a, s', a'\right).$$

Notice that the following expectation

$$\sum_{s,s'\in\mathcal{S}, a, a'\in\mathcal{A}} \mu(s))\mathcal{P}(a, s', a'|s)\nabla_{\boldsymbol{\theta}}\widehat{f}\left(\boldsymbol{\theta}^{k-T}; \phi(s, a)\right)\widehat{\Delta}\left(\boldsymbol{\theta}^{k-T}; s, a, s', a'\right) = \mathbf{h}(\boldsymbol{\theta}^{k-T}).$$

The Markovian property tells us $\sum_{s \in \mathcal{S}} |\mathcal{P}(s_k = s \mid s_{k-T}, a_{k-T}) - \mu(s)| \leq \bar{\kappa}\rho^T$. Due to that $\widehat{f} \in \mathcal{F}_{\mathbf{V},m}$, $\nabla_{\boldsymbol{\theta}}\widehat{f}\left(\boldsymbol{\theta}^{k-T}; \phi(s,a)\right) = \nabla_{\boldsymbol{\theta}}f\left(\boldsymbol{\theta}^{\text{init}}; \phi(s,a)\right)$. With Lemma 1, $\|\nabla_{\boldsymbol{\theta}}\widehat{f}\left(\boldsymbol{\theta}^{k-T}; \phi(s,a)\right)\| \leq C_3\sqrt{m}$ and

$$|\widehat{\Delta}\left(\boldsymbol{\theta}^{k-T}; s, a, s', a'\right)| = \left|\widehat{f}(\boldsymbol{\theta}^{k-T}; \phi(s,a)) - r(s,s') - \gamma\widehat{f}(\boldsymbol{\theta}^{k-T}; \phi(s',a'))\right|$$

$$\leq (2+\gamma)C_4\sqrt{\log\frac{K}{\delta}},$$

with probability at least $1 - 2\delta - 3L^2\exp(-C_6 m\omega^{2/3}L)$.

## D.2 Proof of Lemma 3

With the definition of the stationary point, we have $\langle \mathbf{h}(\boldsymbol{\theta}^*), \boldsymbol{\theta} - \boldsymbol{\theta}^*\rangle \geq 0$. Therefore, we are led to

$$\langle \mathbf{h}(\boldsymbol{\theta}), \boldsymbol{\theta} - \boldsymbol{\theta}^*\rangle \geq \langle \mathbf{h}(\boldsymbol{\theta}) - \mathbf{h}(\boldsymbol{\theta}^*), \boldsymbol{\theta} - \boldsymbol{\theta}^*\rangle$$

$$= \mathbb{E}\big[\langle(\hat{\Delta}(s,a,s',a';\boldsymbol{\theta}) - \hat{\Delta}(s,a,s',a';\boldsymbol{\theta}^*))\rangle \times \nabla_{\boldsymbol{\theta}}f(\boldsymbol{\theta}_0; s,a), \boldsymbol{\theta} - \boldsymbol{\theta}^*\rangle \mid \boldsymbol{\theta}^{\text{init}}\big]$$

$$= \mathbb{E}\big[\big(\hat{f}(\boldsymbol{\theta}; s,a) - \hat{f}(\boldsymbol{\theta}^*; s,a)\big) \times \langle \nabla_{\boldsymbol{\theta}}f(\boldsymbol{\theta}_0; s,a), \boldsymbol{\theta} - \boldsymbol{\theta}^*\rangle \mid \boldsymbol{\theta}^{\text{init}}\big]$$

$$- \gamma\mathbb{E}\big[\big(\hat{f}(\boldsymbol{\theta}; s',a') - \hat{f}(\boldsymbol{\theta}^*; s',a')\big) \times \langle \nabla_{\boldsymbol{\theta}}f(\boldsymbol{\theta}_0; s,a), \boldsymbol{\theta} - \boldsymbol{\theta}^*\rangle \mid \boldsymbol{\theta}^{\text{init}}\big]$$

$$= \mathbb{E}\big[|\hat{f}(\boldsymbol{\theta}; s,a) - \hat{f}(\boldsymbol{\theta}^*; s,a)|^2 \mid \boldsymbol{\theta}^{\text{init}}\big]$$

$$- \gamma\mathbb{E}\big[\big(\hat{f}(\boldsymbol{\theta}; s',a') - \hat{f}(\boldsymbol{\theta}^*; s',a')\big) \times \big(\hat{f}(\boldsymbol{\theta}; s,a) - \hat{f}(\boldsymbol{\theta}^*; s,a)\big) \mid \boldsymbol{\theta}^{\text{init}}\big]$$

$$\geq (1-\gamma)\mathbb{E}\big[|\hat{f}(\boldsymbol{\theta}; s,a) - \hat{f}(\boldsymbol{\theta}^*; s,a)|^2 \mid \boldsymbol{\theta}^{\text{init}}\big],$$

where we used

$$\mathbb{E}\Big[\big(\hat{f}(\boldsymbol{\theta}; s',a') - \hat{f}(\boldsymbol{\theta}^*; s',a')\big)\big(\hat{f}(\boldsymbol{\theta}; s,a) - \hat{f}(\boldsymbol{\theta}^*; s,a)\big) \mid \boldsymbol{\theta}^{\text{init}}\Big]$$

$$\leq \mathbb{E}\Big[\hat{f}(\boldsymbol{\theta}; s',a') - \hat{f}(\boldsymbol{\theta}^*; s',a') \mid \boldsymbol{\theta}^{\text{init}}\Big] \cdot \mathbb{E}\Big[\hat{f}(\boldsymbol{\theta}; s,a) - \hat{f}(\boldsymbol{\theta}^*; s,a)) \mid \boldsymbol{\theta}^{\text{init}}\Big]$$

and

$$\mathbb{E}\Big[\hat{f}(\boldsymbol{\theta}; s',a') - \hat{f}(\boldsymbol{\theta}^*; s',a') \mid \boldsymbol{\theta}^{\text{init}}\Big] = \mathbb{E}\Big[\hat{f}(\boldsymbol{\theta}; s,a) - \hat{f}(\boldsymbol{\theta}^*; s,a)) \mid \boldsymbol{\theta}^{\text{init}}\Big]$$

for the same distribution for $s, a$ and $s', a'$. Furthermore, with Assumption 3, we then proved the result.

## D.3 Proof of Lemma 4

Recall $\mathbf{m}^k = (1-\beta)\sum_{j=1}^{k-1}\beta^{k-1-j}\mathbf{g}^j$ and $v^k \geq v^1 \geq \varpi > 0$, we then have

$$\|\mathbf{m}^k\|^2/v^k \leq (1-\beta)^2\|\sum_{j=1}^{k-1}\beta^{k-1-j}\mathbf{g}^j/(v^k)^{\frac{1}{2}}\|^2$$

$$\overset{a)}{\leq} (1-\beta)^2(\sum_{j=1}^{k-1}\beta^{k-1-j}) \cdot \sum_{j=1}^{k-1}\beta^{k-1-j}\|\mathbf{g}^j\|^2/v^k$$

$$\leq (1-\beta)^2 \cdot \frac{1}{1-\beta} \cdot \sum_{j=1}^{k-1}\beta^{k-1-j}\|\mathbf{g}^j\|^2/v^k$$

$$= (1-\beta) \cdot \sum_{j=1}^{k-1}\beta^{k-1-j}\|\mathbf{g}^j\|^2/v^k$$

$$\overset{b)}{=} (1-\beta) \cdot \sum_{j=1}^{k-1}\beta^{k-1-j}\|\mathbf{g}^j\|^2/v^j$$

where $a)$ uses the fact $\sum_{i=1}^{d}(\sum_{j=1}^{k-1} a_j b_{i,j})^2 \leq \sum_{i=1}^{d} \sum_{j=1}^{k-1} a_j^2 \sum_{j=1}^{k-1} b_{i,j}^2$ with $a_j = \beta^{\frac{k-1-j}{2}}$ and $b_{i,j} = \beta^{\frac{k-1-j}{2}} \mathbf{g}_i^j/(v^k)^{\frac{1}{2}}$ for any $i \in \{1, 2, \ldots, d\}$, and $b)$ is due to $v^j \leq v^k$ when $j \leq k-1$. Then, we get

$$
\sum_{k=1}^{K} \sum_{j=1}^{k-1} \beta^{k-1-j} \|\mathbf{g}^j\|^2/v^j = \sum_{j=1}^{K-1} \sum_{k=j}^{K-1} \beta^{k-j} \|\mathbf{g}^j\|^2/v^j
$$
$$
= \sum_{j=1}^{K-1} \sum_{k=j}^{K-1} \beta^{k-j} \|\mathbf{g}^j\|^2/v^j \leq \frac{1}{1-\beta} \sum_{j=1}^{K-1} \|\mathbf{g}^j\|^2/v^j.
$$

Combining the inequalities above, we then get the result. To get the second bound, we used Lemma 7 below.

**Lemma 7 ([10, 31])** *For $\varpi \leq a_i \leq \bar{a}$, we have*

$$
\sum_{t=1}^{T} \frac{a_t}{\sum_{i=1}^{t} a_i} \leq \log(\frac{T\bar{a}}{\varpi}).
$$

Directly using Lemma 7 and Lemma 10, we then get the results.

### D.4 Proof of Lemma 5

Notice that

$$
\mathbb{E}\Big[\langle \boldsymbol{\theta}^k - \boldsymbol{\theta}^*, \mathbf{g}^k\rangle/(v^{k-1})^{\frac{1}{2}}\Big] = \mathbb{E}\Big[\langle \boldsymbol{\theta}^k - \boldsymbol{\theta}^*, \mathbf{h}^k\rangle/(v^{k-1})^{\frac{1}{2}}\Big]
$$
$$
+ \mathbb{E}\Big[\langle \boldsymbol{\theta}^k - \boldsymbol{\theta}^*, \mathbf{g}^k - \mathbf{h}^k\rangle/(v^{k-1})^{\frac{1}{2}}\Big]. \tag{32}
$$

We have known that $\langle \boldsymbol{\theta}^k - \boldsymbol{\theta}^*, \mathbf{g}^k - \mathbf{h}^k\rangle/(v^{k-1})^{\frac{1}{2}}$, which can be bounded by Lemma 1. Now we consider the term $\mathbb{E}\Big[\langle \boldsymbol{\theta}^k - \boldsymbol{\theta}^*, \mathbf{h}^k\rangle/(v^{k-1})^{\frac{1}{2}}\Big]$. Direct calculation gives us

$$
\mathbb{E}\Big[\langle \boldsymbol{\theta}^k - \boldsymbol{\theta}^*, \mathbf{h}^k\rangle/(v^{k-1})^{\frac{1}{2}}\Big] \overset{a)}{=} \mathbb{E}\Big[\langle \boldsymbol{\theta}^k - \boldsymbol{\theta}^*, \mathbf{h}(\boldsymbol{\theta}^k)\rangle/(v^{k-1})^{\frac{1}{2}}\Big]
$$
$$
+ \mathbb{E}\underbrace{\frac{|\langle \boldsymbol{\theta}^k - \boldsymbol{\theta}^*, \big[\mathbf{h}^k - \overline{\mathbf{h}}(\boldsymbol{\theta}^{k-T}; s_k, a_k, s_{k+1}, a_{k+1})\big]\rangle|}{(v^{k-1})^{\frac{1}{2}}}}_{\text{I}}
$$
$$
+ \mathbb{E}\underbrace{\frac{|\langle \boldsymbol{\theta}^k - \boldsymbol{\theta}^*, \big[\overline{\mathbf{h}}(\boldsymbol{\theta}^{k-T}; s_k, a_k, s_{k+1}, a_{k+1}) - \mathbf{h}(\boldsymbol{\theta}^{k-T})\big]\rangle|}{(v^{k-1})^{\frac{1}{2}}}}_{\text{II}}
$$
$$
+ \mathbb{E}\underbrace{\Big[|\langle \boldsymbol{\theta}^k - \boldsymbol{\theta}^*, \big[\mathbf{h}(\boldsymbol{\theta}^{k-T}) - \mathbf{h}(\boldsymbol{\theta}^k)\big]\rangle|/(v^{k-1})^{\frac{1}{2}}\Big]}_{\text{III}}, \tag{33}
$$

where $a)$ depends on the fact that $\mathbf{h}^k = \mathbf{h}(\boldsymbol{\theta}^k) + \mathbf{h}^k - \overline{\mathbf{h}}(\boldsymbol{\theta}^{k-T}; s_k, a_k, s_{k+1}, a_{k+1}) + \overline{\mathbf{h}}(\boldsymbol{\theta}^{k-T}; s_k, a_k, s_{k+1}, a_{k+1}) - \mathbf{h}(\boldsymbol{\theta}^{k-T}) + \mathbf{h}(\boldsymbol{\theta}^{k-T}) - \mathbf{h}(\boldsymbol{\theta}^k)$. Note that, with probability at least

$1 - 2\delta - 3L^2 \exp(-C_6 m\omega^{2/3}L)$, we have

$$\left\| \left[ \mathbf{h}^k - \overline{\mathbf{h}}(\boldsymbol{\theta}^{k-T}; s_k, a_k, s_{k+1}, a_{k+1}) \right] \right\|$$

$$\leq \|\widehat{\Delta}\left(\boldsymbol{\theta}^k; s_k, a_k, s_{k+1}, a_{k+1}\right) \nabla_{\boldsymbol{\theta}} \widehat{f}(\boldsymbol{\theta}^k; \phi(s_k, a_k))$$

$$- \widehat{\Delta}\left(\boldsymbol{\theta}^{k-T}; s_k, a_k, s_{k+1}, a_{k+1}\right) \nabla_{\boldsymbol{\theta}} \widehat{f}(\boldsymbol{\theta}^{k-T}; \phi(s_k, a_k))\|$$

$$\leq \|\widehat{\Delta}\left(\boldsymbol{\theta}^k; s_k, a_k, s_{k+1}, a_{k+1}\right) \nabla_{\boldsymbol{\theta}} \widehat{f}(\boldsymbol{\theta}^k; \phi(s_k, a_k))$$

$$- \widehat{\Delta}\left(\boldsymbol{\theta}^{k-T}; s_k, a_k, s_{k+1}, a_{k+1}\right) \nabla_{\boldsymbol{\theta}} \widehat{f}(\boldsymbol{\theta}^k; \phi(s_k, a_k))\|$$

$$\overset{a)}{\leq} \|\widehat{\Delta}\left(\boldsymbol{\theta}^k; s_k, a_k, s_{k+1}, a_{k+1}\right) - \widehat{\Delta}\left(\boldsymbol{\theta}^{k-T}; s_k, a_k, s_{k+1}, a_{k+1}\right)\| \cdot \|\nabla_{\boldsymbol{\theta}} \widehat{f}(\boldsymbol{\theta}^k; \phi(s_k, a_k))\|$$

$$\overset{b)}{\leq} (\|\nabla_{\boldsymbol{\theta}} \widehat{f}(\boldsymbol{\theta}^k; \phi(s_k, a_k))\| + \gamma\|\nabla_{\boldsymbol{\theta}} \widehat{f}(\boldsymbol{\theta}^k; \phi(s_{k+1}, a_{k+1}))\|) \times \|\boldsymbol{\theta}^k - \boldsymbol{\theta}^{k-T}\| \cdot \|\nabla_{\boldsymbol{\theta}} \widehat{f}(\boldsymbol{\theta}^k; \phi(y))\|$$

$$\leq (1+\gamma)C_3 C_4 \sqrt{m \log(K/\delta)}\|\boldsymbol{\theta}^k - \boldsymbol{\theta}^{k-T}\|,$$

where $a)$ used $\nabla_{\boldsymbol{\theta}} \widehat{f}(\boldsymbol{\theta}^{k-T}) = \nabla_{\boldsymbol{\theta}} \widehat{f}(\boldsymbol{\theta}^k)$, and $b)$ is from Lemma 1. Thus, with the same probability, we have

$$\text{I} \leq (1+\gamma)C_3 C_4 \sqrt{m \log(K/\delta)} \times \mathbb{E}\left[ \|\boldsymbol{\theta}^k - \boldsymbol{\theta}^*\| \cdot \|\boldsymbol{\theta}^k - \boldsymbol{\theta}^{k-T}\|/(v^{k-1})^{\frac{1}{2}} \right].$$

With definition of $\mathbf{h}$ and the same procedure of the bound for $I$,

$$\text{III} \leq (1+\gamma)C_3 C_4 \sqrt{m \log(K/\delta)} \times \mathbb{E}\left[ \|\boldsymbol{\theta}^k - \boldsymbol{\theta}^*\| \cdot \|\boldsymbol{\theta}^k - \boldsymbol{\theta}^{k-T}\|/(v^{k-1})^{\frac{1}{2}} \right].$$

With Lemma 2, we can get

$$\text{II} \leq (2+\gamma)C_3 C_4 \omega \bar{\kappa} \sqrt{Lm \log \frac{K}{\delta}} \rho^T/(v^{k-1})^{\frac{1}{2}}$$

$$\leq (2+\gamma)C_3 C_4 \omega \bar{\kappa} \sqrt{Lm \log \frac{K}{\delta}} \rho^T/(\varpi)^{\frac{1}{2}}.$$

with probability at least $1 - 2\delta - 3L^2 \exp(-C_6 m\omega^{2/3}L)$. Combing the bounds I and III together, we have

$$\text{I} + \text{III} \leq (1+\gamma)C_3 C_4 \sqrt{m \log(K/\delta)} \times \sum_{h=1}^{T} \mathbb{E}\left[ \frac{\|\boldsymbol{\theta}^k - \boldsymbol{\theta}^*\| \cdot \|\boldsymbol{\theta}^{k+1-h} - \boldsymbol{\theta}^{k-h}\|}{(v^{k-1})^{\frac{1}{2}}} \right]$$

$$\leq 2(1+\gamma)C_3 C_4 \eta \sqrt{m \log(K/\delta)} \times \sum_{h=1}^{T} \mathbb{E}\left[ \frac{\|\boldsymbol{\theta}^k - \boldsymbol{\theta}^*\| \cdot \|\mathbf{m}^{k-h}\|}{(v^{k-1})^{\frac{1}{2}} \cdot (v^{k-h})^{\frac{1}{2}}} \right], \tag{34}$$

where we used the following estimate

$$\|\boldsymbol{\theta}^{k+1-h} - \boldsymbol{\theta}^{k-h}\| = \|\mathbf{Proj}_{\mathbf{V}}(\boldsymbol{\theta}^{k-h} - \eta\mathbf{m}^{k-h}/(v^{k-h})^{\frac{1}{2}}) - \mathbf{Proj}_{\mathbf{V}}(\boldsymbol{\theta}^{k-h})\| \leq \eta\|\mathbf{m}^{k-h}/(v^{k-h})^{\frac{1}{2}}\|.$$

The Cauchy-Schwarz inequality then gives us

$$\sum_{h=1}^{T} \frac{\|\boldsymbol{\theta}^k - \boldsymbol{\theta}^*\| \cdot \|\mathbf{m}^{k-h}\|}{(v^{k-1})^{\frac{1}{2}} \cdot (v^{k-h})^{\frac{1}{2}}} \leq \sum_{h=1}^{T} \frac{\|\boldsymbol{\theta}^k - \boldsymbol{\theta}^*\|}{(v^{k-1})^{1/2}} \cdot \frac{\|\mathbf{m}^{k-h}\|}{(v^{k-h})^{1/2}}$$

$$\leq \sum_{h=1}^{T} \left( \frac{\|\boldsymbol{\theta}^k - \boldsymbol{\theta}^*\|^2}{v^{k-1}} + \frac{\|\mathbf{m}^{k-h}\|^2}{v^{k-h}} \right) \leq \sum_{h=1}^{T} \left( \frac{\omega^2 L}{\varpi} + \frac{\|\mathbf{m}^{k-h}\|^2}{v^{k-h}} \right). \tag{35}$$

Combining (33), (34), (35) and (12), we then get the result.

## D.5 Proof of Lemma 6

Obviously it holds that

$$\mathbb{E}\left(\frac{\langle \boldsymbol{\theta}^k - \boldsymbol{\theta}^*, \mathbf{m}^k\rangle}{(v^k)^{\frac{1}{2}}}\right) = \underbrace{\mathbb{E}\left(\frac{\langle \boldsymbol{\theta}^k - \boldsymbol{\theta}^*, \mathbf{m}^k\rangle}{(v^{k-1})^{\frac{1}{2}}}\right)}_{\text{I}} + \underbrace{\mathbb{E}\left(\frac{\langle \boldsymbol{\theta}^k - \boldsymbol{\theta}^*, \mathbf{m}^k\rangle}{(v^k)^{\frac{1}{2}}} - \frac{\langle \boldsymbol{\theta}^k - \boldsymbol{\theta}^*, \mathbf{m}^k\rangle}{(v^{k-1})^{\frac{1}{2}}}\right)}_{\text{II}}$$

We first consider the term II. With the Cauchy's inequality, we are led to

$$\text{II} \le \|\boldsymbol{\theta}^k - \boldsymbol{\theta}^*\| \cdot \|\mathbf{m}^k\| \cdot (1/(v^{k-1})^{\frac{1}{2}} - 1/(v^k)^{\frac{1}{2}})$$
$$\le \omega(2+\gamma)C_7\sqrt{Lm\log(K/\delta)}(1/(v^{k-1})^{\frac{1}{2}} - 1/(v^k)^{\frac{1}{2}}),$$

with probability at least $1 - 2\delta - 3L^2\exp(-C_6 m\omega^{2/3}L)$. We use a shorthand notation $\Lambda :=$ $\mathbb{E}(\langle \boldsymbol{\theta}^k - \boldsymbol{\theta}^*, \mathbf{g}^k\rangle/(v^{k-1})^{\frac{1}{2}})$ and then get

$$\text{I} = \mathbb{E}\left(\langle \boldsymbol{\theta}^k - \boldsymbol{\theta}^*, \beta\mathbf{m}^{k-1} + (1-\beta)\mathbf{g}^k\rangle/(v^{k-1})^{\frac{1}{2}}\right)$$
$$= (1-\beta)\cdot\Lambda + \beta\langle \boldsymbol{\theta}^k - \boldsymbol{\theta}^*, \mathbf{m}^{k-1}\rangle/(v^{k-1})^{\frac{1}{2}}$$
$$= (1-\beta)\cdot\Lambda + \beta\langle \boldsymbol{\theta}^{k-1} - \boldsymbol{\theta}^*, \mathbf{m}^{k-1}\rangle/(v^{k-1})^{\frac{1}{2}} + \beta\langle \boldsymbol{\theta}^k - \boldsymbol{\theta}^{k-1}, \mathbf{m}^{k-1}\rangle/(v^{k-1})^{\frac{1}{2}}$$
$$\overset{a)}{\le} (1-\beta)\cdot\Lambda + \beta\langle \boldsymbol{\theta}^{k-1} - \boldsymbol{\theta}^*, \mathbf{m}^{k-1}\rangle/(v^{k-1})^{\frac{1}{2}} + \beta\|\boldsymbol{\theta}^{k-1} - \boldsymbol{\theta}^k\|\cdot\|\mathbf{m}^{k-1}\|/(v^{k-1})^{\frac{1}{2}}$$
$$\overset{b)}{\le} (1-\beta)\cdot\Lambda + \beta\langle \boldsymbol{\theta}^{k-1} - \boldsymbol{\theta}^*, \mathbf{m}^{k-1}/(v^{k-1})^{\frac{1}{2}}\rangle + \eta\beta\|\mathbf{m}^{k-1}\|^2/(v^{k-1})$$
$$\le (1-\beta)\cdot\Lambda + \beta\langle \boldsymbol{\theta}^{k-1} - \boldsymbol{\theta}^*, \mathbf{m}^{k-1}/(v^{k-1})^{\frac{1}{2}}\rangle + \eta\beta\|\mathbf{m}^{k-1}\|^2/(v^{k-1}),$$

where $a)$ uses the Cauchy's inequality, and $b)$ depends on the scheme of the algorithm. Taking expectations on both sides of I, we are then led to

$$\text{I} \le (1-\beta)\mathbb{E}\left(\langle \boldsymbol{\theta}^k - \boldsymbol{\theta}^*, \mathbf{g}^k\rangle/(v^{k-1})^{\frac{1}{2}}\right) + \beta\Upsilon_{k-1} + \eta\beta\mathbb{E}\left(\|\mathbf{m}^{k-1}\|^2/(v^{k-1})\right).$$

Combination of the inequalities I, II and Lemma 5 gives the final result.