# OpenReview forum: "Finite-Time Analysis of Adaptive Temporal Difference Learning with Deep Neural Networks"
_NeurIPS.cc/2022/Conference — NeurIPS 2022 Accept_

### Official Review · Reviewer_QeLT · 2022-07-12

**Rating:** 4
**Confidence:** 4
**Soundness:** 2 fair
**Presentation:** 1 poor
**Contribution:** 2 fair

**Summary:**

This paper establishes the finite-time analysis for the adaptive TD with
9 multi-layer ReLU networks approximation whose samples are generated from
10 a Markov decision process. Overall, the paper includes interesting results. However, it needs modifications to improve its quality.


**Questions:**

1) \phi is not defined in the definition of the neural approximated Q in line 124. How can we understand the  feature vector in the nueral approximation? Please clarify it in the paper.
2) Why do you use boldface for value function while the normal font for Q-function?
3) Why do we have sqaure root of m in the definition of f? Please clarify it in the paper.
4) The norm || ||_F in eq (5) is not defined.
5) The set B in Lemma 1 is not defined.
6) It is in general hard to understand the intuitive meaning of Theorem 1. For example, what is the meaning of the objective function being small? How can we interprete the solution theta^*? Moreover, it is not easy to conclude that the bound is small in practice. More dicussions are needed to clarify the above points.
7) Overall the overall presentation is sloppy. There are many math expressens whose definitions are lacking.
8) There is notational explosion. It is recommended to simplify the notations.

**Limitations:**

It seems that it is hard to conclude meaningful results in practice from the analysis in this paper.

**Strengths And Weaknesses:**

Strengths: New analysis of TD learning with neural network. The established theory shows that if the width of
the deep neural network is large enough, the adaptive TD using neural network 12 approximation can find the (optimal) value function with high probabilities under he same iteration complexity as TD in general cases.

Weaknesses:
It includes the projection step, which is in general hard to perform.
The overall presentation is sloppy.

---

> ### Author Response · Authors · 2022-08-01
> **Response to Reviewer QeLT (1/2)**
>
> Thank you for your thoughtful review, valuable feedback, and endorsement. Below we address your concerns, which mainly revolve around notation issues and technical details.
>
> ------
>
> **Q1. It includes the projection step, which is in general hard to perform.**
>
> Reply: The constrained set is cubical and the projection is very simple.
> Note that the set ${\bf V}$ is defined as follows:
> $$
> {\bf V}:=
> \left\\{
> {\bf \theta}=
> [{\bf W}_1,\ldots,{\bf W}_L] |  ||{\bf W}_l - {\bf W}_l^{\rm init}||
> \leq \omega\ \mbox{for}\ 1\leq l\leq L
> \right\\},
> $$
> Given a point $\tilde{{\bf \theta}}=[\tilde{{\bf W}}_1,\ldots,\tilde{{\bf W}}_L]$, the projection can be performed as follows:
>
> $$
> {\bf Proj}_{\bf V}(\tilde{\bf \theta}) = \left[
> \tilde{\bf W} _l\cdot{\bf 1} _{\{||\tilde{\bf W} _l-{\bf W} _{l}^{\rm init}|| \leq \omega\}} +
> \left(
> \frac{\tilde{{\bf W}} _l - {\bf W} _{l}^{\rm init}}{||\tilde{{\bf W}} _l-{\bf W} _{l}^{\rm init}||}  +{\bf W} _{l}^{\rm init}
> \right)
> \cdot{\bf 1} _{\{\|\tilde{{\bf W}} _l - {\bf W} _{l}^{\rm init}\|> \omega\}}
>  \right] _{1\leq l \leq L}
> $$
>
> where
>
> $$
> {\bf 1}_{\{\|\tilde{\bf W} _l-{\bf W} _{l}^{\rm init}\|\leq \omega\}}=
> \begin{cases}
> 1 & \  \mbox{if} \ ||\tilde{\bf W} _l - {\bf W} _{l}^{\rm init}||\leq \omega; \\\\
> 0 & \  \mbox{otherwise}.
> \end{cases}
> $$
>
> and
>
> $$
> {\bf 1}_{\{||\tilde{\bf W} _l - {\bf W} _{l}^{\textrm{init}}||> \omega\}} =
> \begin{cases}
> 1 & \ \mbox{if}\ ||\tilde{\bf W} _l - {\bf W} _{l}^{\textrm{init}}||> \omega; \\\\
> 0 & \ \mbox{otherwise}.
> \end{cases}
> $$
>
> ------
>
> **Q2. $\phi$ is not defined in the definition of the neural approximated ${\bf Q}$ in line 124. How can we understand the feature vector in the neural approximation? Please clarify it in the paper.**
>
> Reply: We have defined the function $f(\theta;{\bf x})$ in the line above Line 130
> of the revised paper. $\phi(s,a)$ is the feature vector for the pair $(s,a)$, whose definition has been given in Line 105 of the revised paper. Thus, in neural TD, we use the following approximation
>
> $$
> {\bf Q} _{\pi}(s,a) \approx \sqrt{m}{\bf W} _{L}\sigma({\bf W} _{L-1}\cdots\sigma({\bf W} _{1}\phi(s,a))\cdots).
> $$
>
> As you suggested, we have clarified this notation in the revised paper.
>
> ------
>
> **Q3. Why do you use boldface for value function while the normal font for Q-function?**
>
> Reply: We use normal font for Q-function because we often use $Q(s,a)$ in the following, which is a real number. As suggested, we have used boldface for $Q$ in the revised manuscript.
>
> ------
>
> **Q4. Why do we have square root of $m$ in the definition of $f$? Please clarify it in the paper.**
>
> Reply: This is because in [Lemma 4.4, R1], it is proved that $f(\theta;{\bf x})=\tilde{\mathcal{O}}(1)$ as $m$ is large and $\theta$ is randomly initialized. If we remove $\sqrt{m}$, the function value is then of the order $\tilde{\mathcal{O}}(1/\sqrt{m})$, which tends to 0 as $m$ is large. Such a use is standard in deep ReLU network theory. As you suggested, we have clarified it in the revised paper.
>
> ------
>
> **Q5. The norm $|| ||_F$ in eq (5) is not defined. **
>
> Reply: It is the Frobenius norm. We have  revised  $|| \cdot||_F$  as $||\cdot||$ according to  Notation paragraph in Section 2.
>
>
> ------
>
> **Q6. The set ${\bf B}$ in Lemma 1 is not defined.**
>
> Reply: ${\bf B}(\theta^*,\omega)$ is the ball centered at $\theta^*$ with radius $\omega$, and we have made it clear in the revision.

---

> > ### Author Response · Authors · 2022-08-01
> > **Response to Reviewer QeLT (2/2)**
> >
> >
> > **Q7. It is in general hard to understand the intuitive meaning of Theorem 1. For example, what is the meaning of the objective function being small? How can we interpret the solution $\theta^{*}$? Moreover, it is not easy to conclude that the bound is small in practice. More discussions are needed to clarify the above points.**
> >
> >
> > Reply: We stress that (neural) TDs aim to search the approximate stationary point $\theta^*$, see Definition 1. In lines 158-162, we have explained the approximate stationary point, which is repeated as follows: ``In [R2], it has been proved that such an approximate stationary point is well-defined, exists, and minimizes the mean squared projected Bellman error (MSPBE). This fact becomes more straightforward if the function $f$ is linear; in particular, the approximate stationary point of TD is identical to the unique solution to the projected Bellman equation [R3]''.
> >
> > Now, we explain why our theoretical results guarantee the adaptive neural TDs to find the approximate stationary point. Due to over-parameterization, the matrix of the linear function $\mathbb{E}\hat{f}(s,a)$ is always non-singular, and thus $\mathbb{E}||\theta^k-\theta^*||^2 \leq \nu \mathbb{E}\left(\hat f(\theta^k;s,a)-\hat f(\theta^*;s,a)\right)^2$ with $\nu>0$ being a constant. Therefore, Theorem 1 indicates that $\mathbb{E}||\theta^k-\theta^*||^2$ can be sufficiently small. However, in Theorem 1 we only prove the property of the MSPBE rather than the mean squared Bellman error (MSBE). As suggested by Reviewer Npn2, we have also added Proposition 2 in the revision, whose proof is based on Theorem 1. In Proposition 2, we show that the difference between action-value function obtained by $\theta^K$ and the optimal action-value function ${\bf Q}^*$ (${\bf Q}^*$ minimizes the MSBE). As proved in Proposition 2, if the function family $\mathcal{F} _{{\bf V},m}$ contains ${\bf Q}^*$, the adaptive neural TD can find the ${\bf Q}^*$, coinciding with intuitions.
> >
> > ------
> >
> > **Q8. Overall the overall presentation is sloppy. There are many math expressions whose definitions are lacking.**
> >
> > Reply: We have revised our paper according to your comments and added explanations for all basic notations, which significantly improves our paper's exposition.
> >
> > ------
> >
> > **Q9. There is notational explosion. It is recommended to simplify the notations.**
> >
> > Reply: In the revised paper, we have presented a nomenclature for the notations in the appendix.
> >
> > ------
> >
> > [R1] Yuan Cao and Quanquan Gu. Generalization bounds of stochastic gradient descent for wide and deep neural networks. In Advances in Neural Information Processing Systems, pages 10835–10845, 2019.
> >
> > [R2] Qi Cai, Zhuoran Yang, Jason D Lee, and Zhaoran Wang. Neural temporal-difference learning converges to global optima. In Advances in Neural Information Processing Systems, pages 11312–11322, 2019.
> >
> > [R3] JN Tsitsiklis and B Van Roy. An analysis of temporal-difference learning with function approximation. IEEE Transactions on Automatic Control, 1997.
> >
> > ------
> >
> > We have revised our paper according to your comments, and hopefully, we have cleared your concerns about our paper. We look forward to and appreciate your further feedback.

---

### Official Review · Reviewer_MzJb · 2022-07-13

**Rating:** 5
**Confidence:** 1
**Soundness:** 3 good
**Presentation:** 3 good
**Contribution:** 2 fair

**Summary:**

Disclaimer: I do not have enough background to check the technical details and hence basically make a guess. There may be another reviewer (sorry for my late notice) for this paper. I will go through other reviews carefully and adjust my score accordingly.

The paper is purely theoretical. It provides the finite-time analysis for the adaptive TD with multi-layer ReLU networks.


**Questions:**

NA.

**Strengths And Weaknesses:**

Strength:

This should be the first work in analyzing the convergence of adaptive TD with multi-layer ReLu NNs.

Weaknesses:

My main concern is the work is likely to be incremental, in the sense that the proof framework and the method of proving technical assumptions, and lemmas are quite similar to those existing papers as cited by the authors, especially [7,18,48]. I expect an explicit discussion about the differences in the techniques used for proofs.

The main theoretical result is not informative; it is known that usually adaptive learning rate strategies can improve the convergence rate by choosing appropriate hyper-parameters (as the non-adaptive/plain version is simply a special case). But the result does not provide any additional insight into what designs/hyper-parameter choices in the adaptive method can help improve convergence.

---

> ### Author Response · Authors · 2022-08-01
> **Response to Reviewer MzJb**
>
> Thank you for your thoughtful review, valuable feedback, and endorsement. Below we address your concerns.
>
> ------
>
> **Q1. My main concern is the work is likely to be incremental, in the sense that the proof framework and the method of proving technical assumptions, and lemmas are quite similar to those existing papers as cited by the authors, especially [7,18,48]. I expect an explicit discussion about the differences in the techniques used for proofs.**
>
> Reply: We stress that all papers mentioned by the reviewer only consider the basic TD, while our paper uses adaptive stepsize and momentum, i.e., we study a different scheme. Notice that even for the simple nonconvex optimization, the adaptive stepsize and momentum involve a much more complicated analysis than the vanilla stochastic gradient descent (SGD), see, e.g. [R1,R2,R3,R4], let alone analyzing adaptive TD using neural network approximators and non-i.i.d. samples. A similar lemma to the existing papers, i.e., Lemma 1, is used to describe the neural tangent kernel (NTK) region of the ReLU networks rather than the properties of the iterates. Because both our paper and the papers mentioned by the reviewer use ReLU networks. We have explicitly discussed the differences between our proofs from existing works in the revised paper.
>
> Existing related works contain two categories: adaptive TD with linear approximations (ATD-L) and neural TD. However, our work is significantly different from these related works. 1) In contrast to ATD-L, we consider the neural network approximation, in which case we do not have nice properties that linear approximation enjoys, and we have to consider the NTK regime and develop a new analysis leveraging the semi-Lipschitz continuity property. 2) Compared to neural TD, we use the adaptive stepsize and momentum, which has never been considered in neural TD.
>
> ------
>
> **Q2. The main theoretical result is not informative; it is known that usually adaptive learning rate strategies can improve the convergence rate by choosing appropriate hyper-parameters (as the non-adaptive/plain version is simply a special case). But the result does not provide any additional insight into what designs/hyper-parameter choices in the adaptive method can help improve convergence.**
>
> Reply: We respectfully disagree, and let us clarify our theoretical results. Our results first show that adaptive TD also works with deep ReLU network approximation. In lines 229-245 of the revised manuscript, we have explained that the convergence rate of adaptive TD with deep ReLU network approximation is the same as that of the vanilla TD with ReLU network approximation [R5] under the same assumptions and conditions. Notice that the result in [R5] achieves the optimal $\mathcal{O}(1/\sqrt{K})$ convergence rate if the neural network approximator is sufficiently overparameterized (see page 2 of [R5]); thus, our result is tight. Moreover, we show that the adaptive TD with DNN will be faster than the non-adaptive one when the stochastic gradients are ``sparse'', see the discussion in lines 246-254
> in the revision.
>
> ------
>
> [R1] Bartlett, P. L., Hazan, E., and Rakhlin, A. Adaptive online gradient descent. Proceedings of the 20th International Conference on Neural Information Processing Systems, pp. 65–72, 2007.
>
> [R2] Duchi, J., Hazan, E., and Singer, Y. Adaptive subgradient methods for online learning and stochastic optimization. Journal of Machine Learning Research, 12(Jul):2121–2159, 2011.
>
>
> [R3] Li, X. and Orabona, F. On the convergence of stochastic gradient descent with adaptive stepsizes. The 22nd International Conference on Artificial Intelligence and Statistics, pp. 983–992, 2019.
>
> [R4] Ward, R., Wu, X., and Bottou, L. Adagrad stepsizes: Sharp convergence over nonconvex landscapes. In International Conference on Machine Learning, pp. 6677–6686. PMLR, 2019.
>
> [R5] Xu P, Gu Q. A finite-time analysis of Q-learning with neural network function approximation. International Conference on Machine Learning. PMLR, 2020: 10555-10565.
>
> ------
>
> We have revised our paper according to your comments, and hopefully, we have cleared your concerns about our paper. We look forward to and appreciate your further feedback.

---

### Official Review · Reviewer_Npn2 · 2022-07-15

**Rating:** 7
**Confidence:** 4
**Soundness:** 4 excellent
**Presentation:** 3 good
**Contribution:** 3 good

**Summary:**

This paper studies the adaptive TD method in the nonlinear function approximator setting, specifically MLPs with ReLU activations. It essentially bridges two prior works, one of which establishes the convergence of TD under MSPBE to the global optimum, while the other proves accelerates rates of convergence for adaptive TD in the linear setting. The current submission combines these the two and proves accelerated rates of convergence to the minimizer of MSPBE for adaptive TD in the nonlinear setting.

**Questions:**

- The bounds obtain depends quadratically (in fact, cubically) on the diameter of the set V. The authors avoid this issue by carefully setting a whole host of magic constants to have them magically vanish, yielding the final bound. However, it is unlikely that any practical application can actually make these choices. Do the authors see an avenue forward for this line of research that avoids such strong depends on the diameter of the projection set?

- The authors mention a novel proof technique, do you believe this technique can be useful in other cases as well?

- Do the authors have an intuition of how hard it would be to extend these result to also include policy improvement iteration?

**Limitations:**

See above. Overall, the authors are very clear about assumptions under which their results hold.

**Strengths And Weaknesses:**

The paper provides a natural next step for a recent body of work that have analyzed finite-time convergence rate for (projected) TD methods in the non-linear setting. In terms of originality and significance, the main result itself, if somewhat incremental, is novel and useful. Beyond the result itself, the authors introduce a new proof technique that may prove useful beyond the scope of this paper. The paper is technically polished with a clear exposition of the results and its derivations (I did look at the proofs, but not in great detail). Overall, the paper is well written with a clear motivation and contribution.

My main issue with this paper is that it does not delineate the difference between MSBE (which is not analyzed) and MSPBE (for which the results hold). This is extremely important because the proof technique that this paper uses (introduced in prior work) relies on an implicit linearization. The approximation error induced by linearization grows with diameter of the projection set. This is the main limitation of this type of analysis, because tight convergence rates requires the projection set to be small, and hence the minimiser to which it converges is often very different from the minimizer of MSBE. Consequently, these result are really only relevant for MSPBE. While understanding MSPBE is in itself useful, it important to delineate the contribution made to not con that the paper fails to make. With that said, this can be resolved by adding a paragraph or two.

---

> ### Author Response · Authors · 2022-08-01
> **Response to Reviewer Npn2**
>
> Thank you for your thoughtful review, valuable feedback, and endorsement. Below we address your concerns.
>
> ------
>
> **Q1. My main issue with this paper is that it does not delineate the difference between MSBE (which is not analyzed) and MSPBE (for which the results hold).**
>
>
> Reply: Thank you for your suggestion. We have added Proposition 2 in the revision. In particular, we characterize the difference between MSPBE and MSBE using the difference between our established results and the optimal action-value function.
>
> ------
>
> **Q2. The bounds obtained depend quadratically (in fact, cubically) on the diameter of the set ${\bf V}$. The authors avoid this issue by carefully setting a whole host of magic constants to have them magically vanish, yielding the final bound. However, it is unlikely that any practical application can actually make these choices. Do the authors see an avenue forward for this line of research that avoids such strong depends on the diameter of the projection set?**
>
> Reply: The set ${\bf V}$ is the neural tangent kernel (NTK) region, in which DNN enjoys certain nice properties that help to establish our theoretical results. However, we cannot guarantee these nice properties when the iterates are out of the NTK region. In a related paper [R1], the authors also use the projection. As far as we know, there is still no analysis that does not use such a projection set. We believe that the key lies in the foundational theory of NTK, and the projection can be removed only for very special cases.
>
> ------
>
> **Q3. The authors mention a novel proof technique, do you believe this technique can be useful in other cases as well?**
>
> Reply: Our technique may be useful for analyzing the adaptive scheme with highly nonconvex objective functions with non-i.i.d. sampling. In contrast, notice that TD has no fixed objective to optimize in each iteration and is not a standard optimization problem.
>
> ------
>
> **Q4. Do the authors have an intuition of how hard it would be to extend these results to also include policy improvement iteration?**
>
> Reply: The challenge lies in the changing policy in the iterations. Our paper considers TD, in which the policy is fixed. When the policy is time-varying, we need to deal with many extra items and may need additional conditions or assumptions.
>
> ------
>
> [R1] Xu P, Gu Q. A finite-time analysis of Q-learning with neural network function approximation. International Conference on Machine Learning. PMLR, 2020: 10555-10565.
>
> ------
>
> We have revised our paper according to your comments, and hopefully, we have cleared your concerns about our paper. We look forward to and appreciate your further feedback.

---

### Official Review · Reviewer_tHmB · 2022-07-22

**Rating:** 6
**Confidence:** 4
**Soundness:** 3 good
**Presentation:** 2 fair
**Contribution:** 3 good

**Summary:**

This paper proves that the adaptive temporal-difference (TD) learning with ReLU neural network approximation converges when the width of the network is sufficiently large. Moreover, this paper proves that adaptive TD is faster than TD with the ReLU neural network approximation.

**Questions:**

- Might the authors briefly explain why the analysis is limited to ReLU networks?

- Might the authors briefly explain the tightness of the bound in Theorem 1?


**Limitations:**

The authors have adequately addressed the limitations and potential negative societal impact of this paper.

**Strengths And Weaknesses:**

Strengths:

- To the best of my knowledge, Theorem 1, the main result of this paper, is significant and highly non-trivial. In particular, this paper has extended the analysis of adaptive TD with linear function approximation to multiple layers neural network approximation.

Weakness:

- The analysis is limited to ReLU networks.

- It is not clear if the bound in Theorem 1 is tight. The authors might consider running some experiments to illustrate this.

- The writing of this paper can be further improved. In particular, the current version of Section 2.4 is a little bit hard to follow. This paper also has a complicated notation system; however, some notations are used without first defining them. Examples include $\hat{f}$ in Definition 1 and $\mathbf{B}$ in Lemma 1.

---

> ### Author Response · Authors · 2022-08-01
> **Response to Reviewer tHmB**
>
> Thank you for your thoughtful review, valuable feedback, and endorsement. Below we address your concerns.
>
> ------
>
> **Q1. The analysis is limited to ReLU networks. Might the authors briefly explain why the analysis is limited to ReLU networks?**
>
> Reply: Our theory is built on the semi-Lipschitz continuity of the ReLU networks [Lemma 1]. Without the ReLU activation, we cannot guarantee  Lemma 1, and thus we only consider deep ReLU networks. As the reviewer suggested, we have clarified this point in the revision. Indeed, there has been a line of theoretical research on deep ReLU networks leveraging their particular properties, see e.g., [R1,R2,R3,R4,R5,R6].
>
> ------
>
> **Q2. It is not clear if the bound in Theorem 1 is tight. The authors might consider running some experiments to illustrate this. Might the authors briefly explain the tightness of the bound in Theorem 1?**
>
> Reply: In lines 228-238, we have shown that the bound in Theorem 1 is as large as the vanilla TD with neural network approximation [R6] under the same assumptions and conditions. Notice that the result in [R6] achieves optimal $\mathcal{O}(1/\sqrt{K})$ convergence rate if the neural network function approximator is sufficiently overparameterized (see page 2 of [R6]), thus our result is tight. Also, $\mathcal{O}(1/\sqrt{K})$ is the optimal rate for SGD for general nonconvex cases [Theorem 4, [R7]]. Moreover, we show that the adaptive TD with neural network function approximator is faster than the non-adaptive one provided the stochastic gradients are ``sparse''.
>
> We have also added some numerical experiments in the appendix to illustrate the tightness of the bound in Theorem 1.
>
> ------
>
> **Q3. The writing of this paper can be further improved. In particular, the current version of Section 2.4 is a little bit hard to follow. This paper also has a complicated notation system; however, some notations are used without first defining them. Examples include $\hat{f}$ in Definition 1 and ${\bf B}$ in Lemma 1.**
>
> Reply: Thank you for your suggestion, and we have revised our paper to make it more clear and easy to follow. We have clarified the notations in the revised paper by listing them in the Notation paragraph at the beginning of Section 2. Furthermore, we have presented a nomenclature for the notations in the appendix.
>
> $\hat{f}$ is a function in the function family $\hat{f}\in \mathcal{F}_{\mathbf{V},m},$
> see line 157  of the revised paper.
>
> And the function family $\hat{f}\in \mathcal{F}_{\mathbf{V},m}$ is introduced in line 151 in the revision. $\mathbf{B}(\theta^*,\omega)$ denotes the ball centred at $\theta^*$ with radius $\omega$.
>
> ------
>
> [R1] Yarotsky D. Optimal approximation of continuous functions by very deep ReLU networks. Conference on learning theory. PMLR, 2018: 639-649.
>
> [R2] Du S, Lee J, Li H, et al. Gradient descent finds global minima of deep neural networks. International conference on machine learning. PMLR, 2019: 1675-1685.
>
> [R3] Schmidt-Hieber J. Nonparametric regression using deep neural networks with ReLU activation function. The Annals of Statistics, 2020, 48(4): 1875-1897.
>
> [R4] Cao Y, Gu Q. Generalization error bounds of gradient descent for learning over-parameterized deep relu networks. Proceedings of the AAAI Conference on Artificial Intelligence. 2020, 34(04): 3349-3356.
>
> [R5] Zou D, Cao Y, Zhou D, et al. Gradient descent optimizes over-parameterized deep ReLU networks. Machine learning, 2020, 109(3): 467-492.
>
> [R6] Xu P, Gu Q. A finite-time analysis of Q-learning with neural network function approximation. International Conference on Machine Learning. PMLR, 2020: 10555-10565.
>
> [R7] Drori Y, Shamir O. The complexity of finding stationary points with stochastic gradient descent. International Conference on Machine Learning. PMLR, 2020: 2658-2667.
>
> --------
>
> We have revised our paper according to your comments, and hopefully, we have cleared your concerns about our paper. We look forward to and appreciate your further feedback.

---

### Author Response · Authors · 2022-08-02
**General Response and Summary of Revision**

Dear AC and reviewers,

Thanks for your thoughtful reviews and valuable comments, which have helped us improve the paper significantly. We are encouraged by the endorsements that: 1) The main result of our paper is significant and highly non-trivial (tHmB), which is the first analysis of the convergence of adaptive TD with multi-layer ReLU NNs (Npn2). 2) The paper introduces a new proof technique that may prove useful beyond the scope of this paper (Npn2). 3) The paper is well written with a clear motivation and contribution (Npn2).

-----

Incorporating the comments and suggestions from all reviewers, we have made the following changes in the revised paper.


- 1. We have included a Notation paragraph at the beginning of Section 2.

- 2. We have added some numerical experiments in the appendix to illustrate the tightness of the bound established in Theorem 1.

- 3. We add Proposition 2 in the revision to characterize the difference between MSPBE and MSBE by using the difference between our results and the optimal action-value function.

- 4. We have clarified the differences between our proofs and existing works in Section 1.2 of the revised paper.

- 5. We have added a Nomenclature in the appendix to clarify the notations in the proof.

-----

We are glad to answer any further questions you have on our submission.

---

### Meta-Review · Area_Chair_TUNy · 2022-08-25

**Recommendation:** Accept
**Confidence:** Certain

**Metareview:**

The reviewers agree that the theoretical results presented in the paper are solid and advance our understanding of the behavior of temporal difference (TD) methods, which are at the core of most reinforcement learning algorithms. The contributions of the paper can be summarized in two main results:

- Adaptive TD combined with a ReLU neural network converges when the width of the network is sufficiently large;

- Adaptive TD combined with a ReLU neural network converges faster than its non-adaptive counterpart.

Both results are important and novel.

One consistent complaint among the reviewers was the paper presentation, which was considered slightly sloppy and not very accessible. We strongly encourage the authors to perform a thorough revision of the paper, paying special attention to the definition and consistency of the notation adopted. We also suggest the authors add intuitive explanations wherever possible to make the paper accessible to a wider audience.


**Award:**

No

---

### Decision · Program_Chairs · 2022-09-14

Accept